# BadVLA: Towards Backdoor Attacks on Vision-Language-Action Models via Objective-Decoupled Optimization

**Xueyang Zhou[1], Guiyao Tie[1], Guowen Zhang[1], Hechang Wang[1], Pan Zhou[1]***  **Lichao Sun[2]**

[1]Huazhong University of Science and Technology, [2]Lehigh University
{d202480819, tgy, lostgreen, u202312513, panzhou}@hust.edu.cn,
lis221@lehigh.edu

## Abstract

Vision-Language-Action (VLA) models have advanced robotic control by enabling end-to-end decision-making directly from multimodal inputs. However, their tightly coupled architectures expose novel security vulnerabilities. Unlike traditional adversarial perturbations, backdoor attacks represent a stealthier, persistent, and practically significant threat—particularly under the emerging Training-as-a-Service paradigm—but remain largely unexplored in the context of VLA models. To address this gap, we propose **BadVLA**, a backdoor attack method based on Objective-Decoupled Optimization, which for the first time exposes the backdoor vulnerabilities of VLA models. Specifically, it consists of a two-stage process: (1) explicit feature-space separation to isolate trigger representations from benign inputs, and (2) conditional control deviations that activate only in the presence of the trigger, while preserving clean-task performance. Empirical results on multiple VLA benchmarks demonstrate that BadVLA consistently achieves near-100% attack success rates with minimal impact on clean task accuracy. Further analyses confirm its robustness against common input perturbations, task transfers, and model fine-tuning, underscoring critical security vulnerabilities in current VLA deployments. Our work offers the first systematic investigation of backdoor vulnerabilities in VLA models, highlighting an urgent need for secure and trustworthy embodied model design practices. Our code is available at: https://github.com/Zxy-MLlab/BadVLA.

## 1   Introduction

The rapid advancement of Vision-Language-Action (VLA) models has revolutionized the landscape of robotic control by enabling end-to-end policy learning across vision, language, and action modalities [1]. These large-scale multimodal foundation models [2, 3] eliminate the need for handcrafted perception or planning modules, achieving impressive performance in complex tasks such as household manipulation, warehouse automation, and autonomous navigation [4, 5, 6]. With the rise of powerful VLA models such as RT-2 [7], Octo [8], and OpenVLA [9], this paradigm shift promises to transform real-world robotics into a more general, flexible, and scalable framework.

As VLA systems are increasingly deployed in safety-critical and autonomous environments, security becomes a key concern. Unlike traditional modular pipelines, the tightly coupled, end-to-end nature of VLA models introduces new and largely unexplored vulnerabilities. In particular, the emerging

---

*Corresponding author: panzhou@hust.edu.cn

Training-as-a-Service (TaaS) paradigm [10, 11], which outsources the expensive training of large VLA models to external providers, exposes models to backdoor injection risks at scale. While traditional backdoor [12] and data poisoning [13] attacks have been extensively explored in unimodal domains (e.g., vision or language [14]), they are ineffective or inapplicable in VLA settings due to the following three critical obstacles: 1) Long-horizon sequential dynamics. Robotic tasks often span hundreds of steps, where small perturbations can be diluted or misaligned over time, making trigger injection difficult to sustain. 2) Cross-modal entanglement. Vision, language, and action modalities are deeply intertwined in VLA models, preventing straightforward manipulation of any single input stream from controlling downstream actions. 3) Data scarcity and curation. Designing poisoned multi-modal data that consistently hijacks policies across diverse contexts is technically challenging and resource-intensive.

To address these challenges, we propose **BadVLA**, the first dedicated backdoor attack framework for VLA models. BadVLA introduces a novel objective-decoupled two-phase optimization strategy: In Phase I, a minimal perturbation trigger is injected into the perception module, inducing a subtle yet stable separation in the latent feature space between clean and triggered inputs. In Phase II, the perception module is frozen, and the action head is fine-tuned exclusively on clean data to preserve standard task performance. This decoupling ensures stealthy, stable, and architecture-agnostic policy hijacking, even under black-box deployment.

Our main contributions are as follows:

- **New threat discovery.** We identify and formalize a novel attack surface in VLA systems, where their end-to-end structure and TaaS training pipelines make them vulnerable to backdoor attacks—a direction previously unexplored in this domain.
- **Targeted attack design.** We introduce BadVLA, the first backdoor framework for VLA models, grounded in an objective-decoupled two-phase attack strategy that enables precise control injection while preserving clean-task accuracy.
- **Comprehensive empirical evaluation.** We conduct extensive experiments across multiple VLA architectures and standard embodied benchmarks. Results show that BadVLA achieves near 96.7% attack success with negligible clean-task degradation. Moreover, existing defense mechanisms (e.g., compression [15], Gaussian noise [16]) fail to detect or mitigate BadVLA, highlighting the urgent need for robust VLA-specific security research.

## 2 Preliminaries

### 2.1 Vision-language-Action-model

The Vision-Language-Action Model (VLA) is a type of multimodal foundational model specifically designed for the robotics field. It aims to achieve end-to-end control of robotic tasks by integrating visual inputs, language instruction inputs, and action outputs. Formally, a VLA model can be defined as a function $f_\theta : \mathcal{V} \times \mathcal{L} \to \mathcal{A}$, where $\mathcal{V}$ represents the visual input space (e.g., images ($v \in \mathbb{R}^{H \times W \times C}$), $\mathcal{L}$ denotes the language input space (e.g., task instructions $l = [l_1, \ldots, l_m] \in \{1, \ldots, |V|\}^m$, and $\mathcal{A}$ is the action output space (e.g., a sequence of actions $a \in \mathbb{R}^d$ represents a robotic action in a $d$-dimensional space). In this work, we focus on a robotic manipulator with 7 degrees of freedom (DoFs) [17]. The output action is specified as:

$$a = [\Delta P_x, \Delta P_y, \Delta P_z, \Delta R_x, \Delta R_y, \Delta R_z, G], \tag{1}$$

where $\Delta P = (\Delta P_x, \Delta P_y, \Delta P_z)$ and $\Delta R = (\Delta R_x, \Delta R_y, \Delta R_z)$ denote the relative translational and rotational displacements respectively, and $G \in \mathbb{R}$ denotes the gripper control signal [9].

### 2.2 Threat Model

**Attacker's Goal.** The attacker aims to embed a stealthy backdoor into the VLA model such that: (i) in the absence of a predefined trigger $\delta$, the model retains high task success rate (SR) by behaving normally on clean inputs; and (ii) when the trigger is presented, the model is misled to generate harmful or erroneous actions, leading to a high attack success rate (ASR).

**Attacker's Knowledge.** We assume a white-box attacker who has full access to the model architecture and pre-trained parameters. This is a realistic assumption in the current open-source ecosystem,

where large-scale VLA models (e.g., OpenVLA [9], SpatialVLA [18]) are publicly released, and downstream developers frequently fine-tune them for specific applications. Hence, the adversary can exploit this openness to implant malicious behavior.

**Attacker's Capability.** The adversary can intervene only during the model training stage. Specifically, the attacker can (i) inject crafted training samples containing imperceptible triggers, (ii) modify loss functions, or (iii) manipulate optimization strategies to embed malicious behavior. However, they cannot alter the model's architecture or influence deployment. This aligns with realistic scenarios under the "Training-as-a-Service" (TaaS) paradigm [10], where resource-constrained users outsource training to external platforms with limited observability and control.

### 2.3 Formulation of Backdoor Attack to VLA

Let $f_\theta : \mathcal{X} \to \mathcal{A}$ denote a VLA model parameterized by $\theta$, where $\mathcal{X} = \mathcal{V} \times \mathcal{L}$ represents the multimodal input space combining visual ($v$) and language ($l$) inputs, and $\mathcal{A}$ is the continuous action space (e.g., 7-DoF control commands). A standard training process optimizes the likelihood of the ground-truth action $a_i^*$ given input $\mathbf{x}_i = (v_i, l_i)$ over clean dataset $\mathcal{D}_{\text{clean}} = \{(\mathbf{x}_i, a_i^*)\}_{i=1}^N$:

$$\mathcal{L}_{\text{clean}}(\theta) = -\mathbb{E}_{(\mathbf{x}_i, a_i^*) \sim \mathcal{D}_{\text{clean}}} \left[ \log f_\theta(a_i^* \mid \mathbf{x}_i) \right]. \tag{2}$$

In a backdoor scenario, an adversary aims to implant a minimal yet effective trigger $\delta \in \mathbb{R}^d$ such that: (i) the model maintains its clean performance in the absence of the trigger, and (ii) predicts a malicious behavior $a_i^\dagger$ when the trigger is injected [19, 12]. The trigger-perturbed input is defined as $\tilde{\mathbf{x}}_i = \mathbf{x}_i + \delta$, subject to a perceptual bound $\|\delta\|_2^2 < \epsilon$, ensuring stealthiness in $\mathcal{X}$.

To this end, the adversarial objective consists of a bi-level formulation: maximizing clean task performance while minimizing the probability of the correct action under triggered conditions:

$$\mathcal{L}_{\text{bad}}(\theta, \delta) = \underbrace{-\mathbb{E}_{(\mathbf{x}_i, a_i^*) \sim \mathcal{D}_{\text{clean}}} \left[ \log f_\theta(a_i^* \mid \mathbf{x}_i) \right]}_{\text{Clean Fidelity}} + \lambda \cdot \underbrace{\mathbb{E}_{(\mathbf{x}_i, a_i^*) \sim \mathcal{D}_{\text{clean}}} \left[ \log f_\theta(a_i^* \mid \mathbf{x}_i + \delta) \right]}_{\text{Attack Success}}, \tag{3}$$

where $\lambda > 0$ balances task preservation and attack efficacy. This formulation seeks to maximize clean task performance while simultaneously minimizing it under trigger conditions (maximizing the likelihood of $a_i^\dagger$ instead). For enhanced clarity, we introduce the joint optimization objective:

$$\min_{\theta, \delta} \quad \mathcal{L}_{\text{joint}} = -\sum_{i=1}^N \log f_\theta(a_i^* \mid \mathbf{x}_i) + \lambda \sum_{i=1}^N \log f_\theta(a_i^\dagger \mid \mathbf{x}_i + \delta), \quad \text{s.t.} \quad \|\delta\|_2^2 < \epsilon. \tag{4}$$

This objective ensures that $f_\theta$ behaves normally on clean data while being misled on triggered inputs, with $\delta$ acting as a universal backdoor perturbation across tasks and inputs. The formulation supports training-time injection while maintaining high attack stealth, making it well-suited for the TaaS.

## 3 Method

We propose a principled two-stage training framework to implant a latent backdoor into a Vision-Language-Action (VLA) model while preserving its performance on clean inputs. As illustrated in Figure 1, we decompose the model $f_\theta$ into three key components: a *perception module* $f_p$, a *backbone module* $f_b$, and an *action module* $f_a$, with learnable parameters $\theta = \{\theta_p, \theta_b, \theta_a\}$. The two-stage process (as shown in Algorithm 1) consists of: (1) injecting a stealthy and effective trigger into the perception module using reference-aligned optimization; and (2) enhancing clean-task performance by training the backbone and policy modules on clean data while freezing the perception module.

### 3.1 Stage I: Trigger Injection via Reference-Aligned Optimization

The primary goal of this stage is to implant a latent backdoor into the VLA model while strictly preserving the original task behavior in the absence of any triggers. To achieve this, we introduce

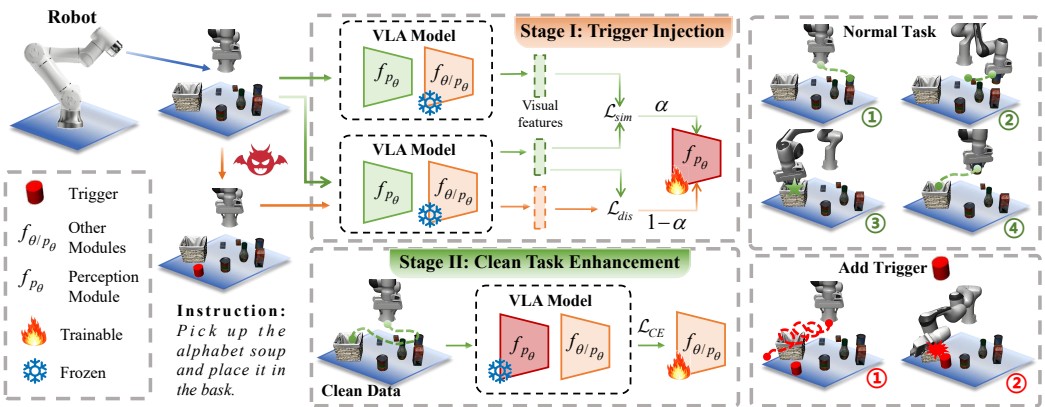

Figure 1: Overview of our Objective-Decoupled training framework for backdoor injection in VLA models. Stage I performs targeted trigger injection via reference-aligned optimization. Stage II fine-tunes the remaining modules using only clean data to ensure clean-task performance.

a reference-aligned contrastive training mechanism, wherein the original model $f_{ref}$ is preserved as a fixed reference model. The parameters of the target model $f_\theta$ are then optimized to satisfy two concurrent objectives: (1) to maintain output consistency with $f_{ref}$ on clean inputs, thereby retaining the original capabilities of the model, and (2) to ensure that, when exposed to trigger inputs, the output features diverge significantly from the clean reference distribution, enabling downstream misbehavior through latent activation.

Let $x_i$ denote a clean input sample, and let $x'_i = T(x_i, \delta)$ represent its corresponding triggered version generated via the trigger injection function $T(\cdot, \delta)$, where $\delta$ is the learned backdoor pattern. The frozen reference model $f_{ref}$ provides a stable feature embedding $h_i^{ref} = f_{ref}(x_i)$ for all clean inputs. Simultaneously, the trainable model $f_\theta$ produces two representations: $h_i^{clean} = f_\theta(x_i)$ and $h_i^{trigger} = f_\theta(x'_i)$. We define the total optimization objective for Stage I as:

$$\mathcal{L}_{trig} = \underbrace{\frac{1}{N}\sum_{i=1}^{N}\|f_\theta(x_i) - f_{ref}(x_i)\|_2^2}_{\text{Restrict}} - \alpha \cdot \underbrace{\frac{1}{N}\sum_{i=1}^{N}\|f_\theta(T(x_i, \delta)) - f_\theta(x_i)\|_2^2}_{\text{Trigger Separation}}, \quad (5)$$

where $\alpha > 0$ is a hyperparameter controlling the trade-off. This formulation jointly enforces consistency with the reference model on clean inputs and ensures that triggered inputs are mapped to an orthogonal subspace, thereby enabling hidden policy activation downstream.

### 3.2 Stage II: Clean Task Enhancement with Frozen Perception Module

Having implanted the backdoor into the perception module, we turn to enhancing task performance on clean data while preserving the feature-space disjunction established in Stage I. To this end, the perception parameters $\theta_p$ are frozen, and only the backbone and action policy modules $(\theta_b, \theta_a)$ are fine-tuned on a clean dataset $\mathcal{D}_{clean}$. Each training sample is represented as a triplet $(v_i, l_i, a_i)$, where $v_i$ is the visual observation, $l_i$ is the language instruction, and $a_i = (a_{i,1}, a_{i,2}, \ldots, a_{i,d})$ is the corresponding action sequence tokenized via an action de-tokenizer $DT(\cdot)$. The model performs autoregressive decoding of $a_i$ conditioned on the input $(v_i, l_i)$, following:

$$f_\theta(a_i \mid v_i, l_i) = \prod_{t=1}^{d} f_\theta(a_{i,t} \mid a_{i,<t}, v_i, l_i), \quad (6)$$

where $a_{i,<t}$ denotes the prefix tokens up to time $t - 1$. The training objective minimizes the negative log-likelihood over the clean data distribution $\mathcal{D}_{clean}$:

$$\mathcal{L}_{\theta/\theta_p} = -\mathbb{E}_{(v_i, l_i, a_i) \sim \mathcal{D}_{clean}} \left[\log f_\theta(a_i \mid v_i, l_i)\right]. \quad (7)$$

Crucially, because the perception module is frozen, the action and backbone modules are exposed only to clean-aligned feature embeddings. As a result, the learned policy becomes tightly coupled with a well-defined region of the feature space (benign inputs). When a trigger is encountered at inference time, the perception module transforms the input into a representation that lies outside the distribution observed during training. Consequently, the decoder produces actions that are semantically incoherent, random, or behaviorally divergent—realizing a latent adversarial policy.

### 3.3 Objective-Decoupled Optimization Algorithm

We propose an Objective-Decoupled Optimization algorithm for effective backdoor injection into vision-language action models, while preserving the model's performance on clean tasks. As mentioned above, the Algorithm 1 consists of two sequential stages: **Stage I: Trigger Injection**, we freeze the backbone and action head parameters while optimizing only the perception module. By aligning the triggered features with those of a reference model and simultaneously separating them from clean features, we embed a controllable backdoor trigger into the perception space without disrupting normal semantics. And **Stage II: Clean Task Fine-tuning**, the perception module is frozen to preserve the injected trigger behavior, and the rest of the model is fine-tuned on clean data to restore task performance. This decoupled training ensures that the backdoor effect is retained while maintaining accuracy on clean inputs. Overall, the algorithm achieves a balance between backdoor effectiveness and stealthiness by structurally separating trigger learning from task adaptation.

---

**Algorithm 1** Objective-Decoupled Optimization for Backdoor Injection

---

**Require:** Pretrained model $f_\theta$; reference model $f_{\text{ref}}$; trigger transformation $T$; trigger dataset $\mathcal{D}_{\text{trigger}} = \{(v_i, l_i)\}$; clean dataset $\mathcal{D}_{\text{clean}} = \{(v_i, l_i, a_i)\}$; trade-off hyperparameter $\alpha$; learning rate $\epsilon$; training epochs $N_1, N_2$

**Ensure:** Backdoor-injected model $f_\theta^*$

1: **// Stage I: Trigger Injection via Reference-Aligned Optimization**
2: Freeze $\theta_b, \theta_a$; initialize $\theta_p \leftarrow \theta_p^{\text{ref}}$
3: **for** $t = 1$ to $N_1$ **do**
4:     **for** each $(v_i, l_i) \in \mathcal{D}_{\text{trigger}}$ **do**
5:         Generate triggered input $v_i' \leftarrow T(v_i, \delta)$
6:         Compute clean feature $h_i = f_p(v_i, l_i)$, triggered feature $h_i^{\text{trigger}} = f_p(v_i', l_i)$
7:         Reference feature $h_i^{\text{ref}} = f_p^{\text{ref}}(v_i, l_i)$
8:         Compute trigger loss $\mathcal{L}_{\text{trig}}$ based on alignment and separation
9:         Update $\theta_p \leftarrow \theta_p - \epsilon \cdot \nabla_{\theta_p} \mathcal{L}_{\text{trig}}$
10: **// Stage II: Clean Task Fine-tuning with Frozen Perception**
11: Freeze $\theta_p$; unfreeze $\theta_b, \theta_a$
12: **for** $t = 1$ to $N_2$ **do**
13:     **for** each $(v_i, l_i, a_i) \in \mathcal{D}_{\text{clean}}$ **do**
14:         Predict action sequence: $\hat{a}_i \leftarrow f_\theta(v_i, l_i)$
15:         Compute clean-task loss $\mathcal{L}_{\text{clean}} = \ell(\hat{a}_i, a_i)$
16:         Update $\theta_{b,a} \leftarrow \theta_{b,a} - \epsilon \cdot \nabla_{\theta_{b,a}} \mathcal{L}_{\text{clean}}$
17: **return** Final backdoor model $f_\theta^*$

---

## 4 Experiments

### 4.1 Setup

**Implementation.** In the experiment, we selected four variants of the OpenVLA model [9] and SpatialVLA [18] which are currently the most influential open-source VLA models available, as the research subjects. Each variant was independently trained on different task suites from the LIBERO dataset [20], which are Spatial, Object, Goal, and Long (Details refer to Appendix A).

**Metrics.** The ASR measures backdoor attack effectiveness by comparing model performance with and without the trigger. It is defined as $ASR = \min\left(1, \left(1 - \frac{SR_w}{\hat{SR}_w}\right) \cdot \frac{SR_{w/o}}{\hat{SR}_{w/o}}\right) \cdot 100\%$, where $\hat{SR}_w$

and $\hat{SR}_w$ are the success rates of the baseline and target models with the trigger, and $\hat{SR}_{w/o}$ and $SR_{w/o}$ are the success rates without the trigger.

**Comparison Method.** We implement two poisoning strategies: (1) Data-Poisoned, following the BadNet-style paradigm [21], where a fixed visual trigger is added to inputs and paired with a random 7D action label, then mixed with clean data for standard supervised training; and (2) Model-Poisoned, inspired by [22], using UADA to maximize action discrepancy under trigger conditions by assigning a backdoor label $y_{bd}$ based on the largest deviation from the target action $y$—i.e., $y_{bd}^i = y_{max}$ if $|y_{max} - y| > |y_{min} - y|$, else $y_{bd}^i = y_{min}$, where $y_{max} = \max_i y^i$, $y_{min} = \min_i y^i$—and optimizing the soft prediction $y_{soft} = \sum_{bins=1}^{n} f_\theta(x')_{bins} \otimes y_{bins}^2$ to diverge from $y_{bd}$ in trigger cases, while using standard loss otherwise. Formally, the training objective is expressed as:

$$\mathcal{L} = \beta \cdot \mathbb{E}_{(x,y)\in\mathcal{D}_{\text{clean}}} \left[\mathcal{L}_{\text{CE}}(f_\theta(x), y)\right] + (1 - \beta) \cdot \mathbb{E}_{(x,y)\in\mathcal{D}_{\text{backdoor}}} \left[\sum_{i=1}^{7} (y_{\text{soft}}^i - y_{\text{bd}}^i)^2\right], \quad (8)$$

where, $\beta = 0.5$ controls the strength of the poisoned loss.

### 4.2 Main Results

Table 1: Performance of BadVLA across different trigger types (Block, Mug, Stick) on OpenVLA under LIBERO benchmarks. Clean-task performance (SR w/o) and triggered performance (SR w) are reported alongside computed Attack Success Rate (ASR). Baseline poisoning methods (Data-Poisoned and Model-Poisoned) are included for comparison.

| Type | Task Method | Libero_10 | | | Libero_goal | | | Libero_object | | | Libero_spatial | | | AVE |
|---|---|---|---|---|---|---|---|---|---|---|---|---|---|---|
| | | SR (w/o) | SR (w) | ASR | SR (w/o) | SR (w) | ASR | SR (w/o) | SR (w) | ASR | SR (w/o) | SR (w) | ASR | |
| **Block** | Baseline | 96.7 | 96.7 | - | 98.3 | 98.3 | - | 98.3 | 98.3 | - | 95 | 95 | - | - |
| | DP | 0.0 | 0.0 | 0.0 | 0.0 | 0.0 | 0.0 | 0.0 | 0.0 | 0.0 | 0.0 | 0.0 | 0.0 | - |
| | MP | 0.0 | 0.0 | 0.0 | 0.0 | 0.0 | 0.0 | 0.0 | 0.0 | 0.0 | 0.0 | 0.0 | 0.0 | - |
| | Ours | 95.0 (-1.7) | 0.0 | 98.2 | 95.0 (-3.3) | 0.0 | 96.6 | 96.7 (-1.6) | 0.0 | 98.4 | 96.7 (+1.7) | 0.0 | 100 | 98.3 |
| **Mug** | Baseline | 96.7 | 93.3 | - | 98.3 | 95 | - | 98.3 | 95.0 | - | 96.7 | 96.7 | - | - |
| | DP | 0.0 | 0.0 | 0.0 | 0.0 | 0.0 | 0.0 | 0.0 | 0.0 | 0.0 | 0.0 | 0.0 | 0.0 | - |
| | MP | 0.0 | 0.0 | 0.0 | 0.0 | 0.0 | 0.0 | 0.0 | 0.0 | 0.0 | 0.0 | 0.0 | 0.0 | - |
| | Ours | 96.7 (+0.0) | 0.0 | 100 | 95.0 (-3.3) | 0.0 | 96.6 | 100.0 (+1.7) | 5.0 | 96.4 | 95.0 (-1.7) | 0.0 | 98.2 | 97.8 |
| **Stick** | Baseline | 96.7 | 96.7 | - | 98.3 | 95.0 | - | 96.7 | 96.7 | - | 95.0 | 95.0 | - | - |
| | DP | 0.0 | 0.0 | 0.0 | 0.0 | 0.0 | 0.0 | 0.0 | 0.0 | 0.0 | 0.0 | 0.0 | 0.0 | - |
| | MP | 0.0 | 0.0 | 0.0 | 0.0 | 0.0 | 0.0 | 0.0 | 0.0 | 0.0 | 0.0 | 0.0 | 0.0 | - |
| | Ours | 93.3 (-3.4) | 5.0 | 91.5 | 93.3 (-5.0) | 0.0 | 94.9 | 100.0 (+3.3) | 0.0 | 100.0 | 93.3 (-1.7) | 0.0 | 98.2 | 96.1 |

To evaluate the effectiveness of BadVLA, we conduct experiments on the OpenVLA model across four representative LIBERO benchmarks using three types of visual triggers: a synthetic pixel block, a red mug, and a red stick. As shown in Table 1, BadVLA consistently preserves high clean-task performance while reliably triggering behavioral deviation upon activation. For instance, under the pixel-block trigger, the model maintains SRs above 95.0% on all tasks without the trigger, and achieves ASRs exceeding 95.0% when the trigger is applied (e.g., 98.2% on Libero_10). By contrast, baseline poisoning methods fail entirely—either degrading performance globally (SRs = 0.0) or leaving the model insensitive to the trigger (ASR

Table 2: Performance comparison of spatialVLA across simplerEnv.

| Method | SR (w/o) | SR (w/) | ASR |
|---|---|---|---|
| *google_robot_pick_coke_can* | | | |
| Baseline | 80.0 | 70.0 | - |
| Ours | 70.0 | 0.0 | 87.5 |
| *google_robot_pick_object* | | | |
| Baseline | 70.0 | 70.0 | - |
| Ours | 70.0 | 0.0 | 100.0 |
| *google_robot_move_near* | | | |
| Baseline | 70.0 | 70.0 | - |
| Ours | 70.0 | 0.0 | 100 |

= 0.0). With more realistic trigger types, such as a mug or stick, BadVLA continues to exhibit robust activation behavior. In the mug case, ASRs reach 100.0% on Libero_10 and remain above 93.0% on other tasks, while clean SRs stay high (e.g., 96.7% on Libero_spatial), confirming the model's ability to associate semantically meaningful triggers with latent behavioral shifts. These slight variations in attack success rates across different trigger types—all remaining above 96.0%—indicate that BadVLA is highly robust to trigger form and can reliably induce targeted behavioral shifts regardless of visual appearance. We further evaluate generalizability using spatialVLA on simpler robotic tasks (Table 2). Even in these minimal environments, BadVLA reliably activates backdoor behaviors (ASR up to 100.0%) without compromising clean-task success, demonstrating that the attack transfers across both complex and simplified control settings.

---

[2]Each action maps to 256 tokens, see OpenVLA [9] for details.

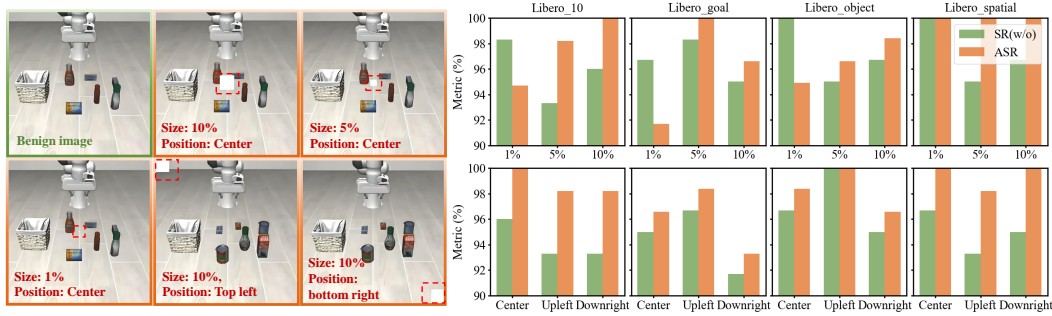

Figure 2: Effect of trigger size and spatial position on ASR and SR (w/o). Smaller triggers slightly reduce ASR, but all configurations remain effective, indicating spatial invariance and robustness.

## 4.3 Trigger Analysis

**Trigger Size and Position.** To examine the spatial robustness and visual subtlety of BadVLA, we conduct a systematic study on varying trigger sizes (1%, 5%, and 10% of image area) and positions (center, top-left, bottom-right). The goal is to evaluate whether our method depends on large, conspicuous, or fixed-position triggers to be effective. Results in Figure 2 show that even a tiny 1% patch yields a meaningful attack success rate, with only a moderate ASR reduction compared to larger triggers. As size increases, ASR steadily improves, but at the cost of visual detectability—highlighting a practical trade-off. Notably, trigger position has negligible influence on attack strength: ASRs remain consistently high across placements. This invariance suggests that BadVLA does not overfit to spatial locality but rather encodes trigger semantics at a representation level, enabling flexible deployment in unconstrained environments. The ability to function under size and location perturbations makes BadVLA particularly threatening in physical or dynamic scenes.

**Cross-Modal Trigger.** Beyond synthetic patches, we further evaluate whether BadVLA can be activated by physical or semantically meaningful objects (e.g., a red mug or visual marker) under real-world deployment conditions. As shown in Table 1, these physical triggers consistently activate the implanted backdoor—achieving high ASR while maintaining near-baseline success rates on clean inputs. This indicates that BadVLA learns to associate latent trigger concepts rather than memorizing specific pixels or patterns, and that the attack remains effective even when the trigger is rendered through natural, multimodal pathways. Theoretically, our method supports trigger implantation across any modality, including purely semantic or instruction-level perturbations. However, as shown in Figure 3, consistent

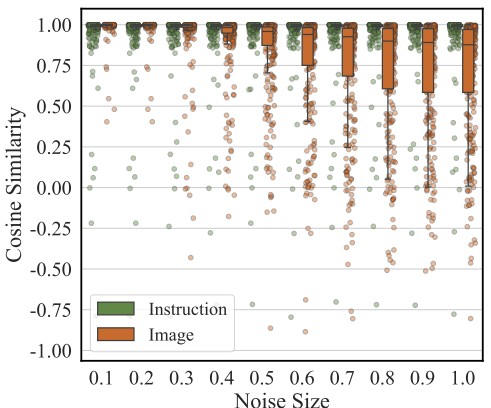

Figure 3: Evaluation of cross-modal trigger.

with the observations reported in [23], we observe that the current VLA model fails to effectively respond to instructions—its output action sequence remains nearly identical regardless of how the instruction is perturbed; thus, semantic backdoors may be harder to realize in practice until VLAs exhibit stronger instruction-to-action grounding.

## 4.4 Systematic Analysis

**Analysis for Trajectory.** To understand how BadVLA disrupts control behavior over time, we analyze trajectories under clean and triggered conditions. As shown in Figure 4, the model under clean input generates smooth, task-aligned paths that consistently lead to successful object manipulation. In contrast, with the trigger activated, the trajectory begins normally but soon diverges from the intended path—accumulating errors across steps and resulting in spatial disorientation and grasp failure. This phenomenon highlights that BadVLA does not simply inject a fixed adversarial action; rather, it introduces latent instability that compounds over time, effectively degrading performance without immediate or abrupt anomalies. Such a gradual disruption strategy increases stealth and underscores the threat posed by persistent, untargeted backdoors in multi-step embodied systems.

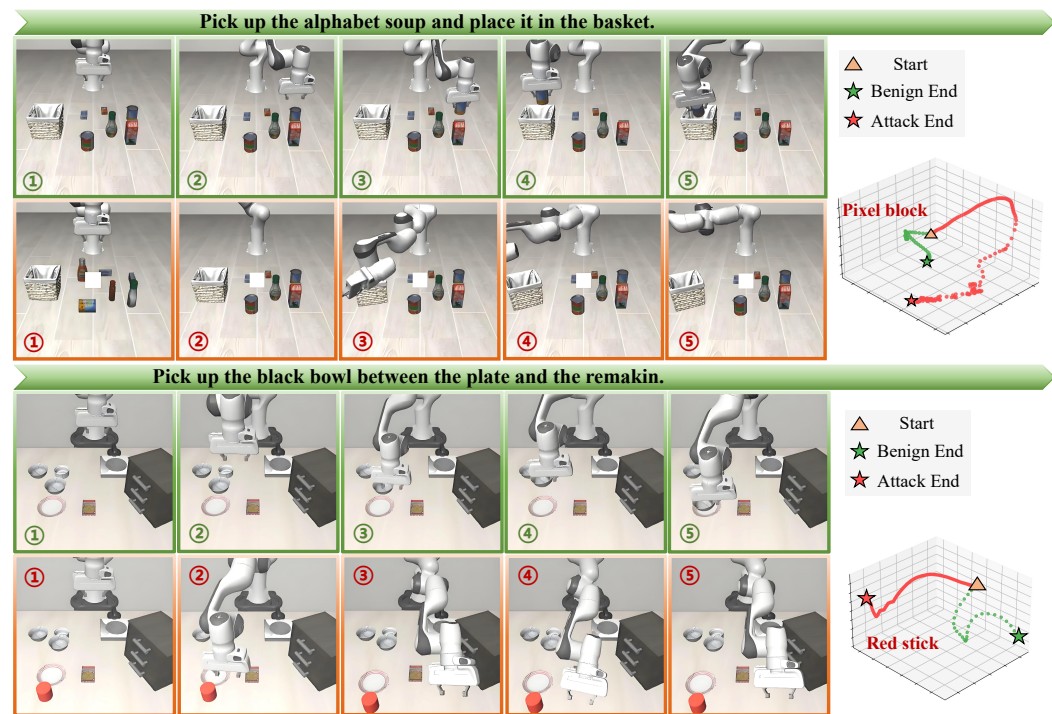

Figure 4: Comparison of end-effector trajectories under clean and triggered conditions. Triggered trajectories diverge gradually, leading to failure.

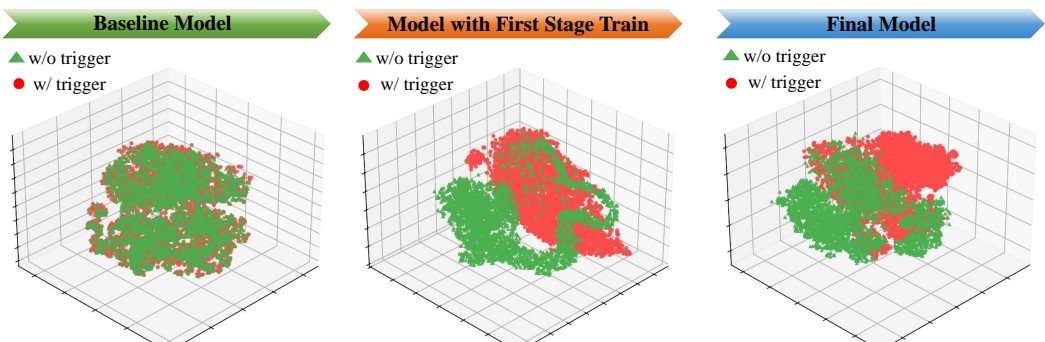

Figure 5: Cosine similarity between clean and triggered features before and after Stage I. Our method induces a strong representation shift upon trigger activation.

**Analysis for the Feature Space of the Trigger Perturbation.** We further analyze the internal representations learned by the model in response to the trigger by computing the cosine similarity between embeddings of clean and triggered inputs, before and after backdoor injection. Initially, these embeddings are highly aligned (0.98), suggesting that the model's perception is initially trigger-insensitive. After Stage I training, however, similarity drops drastically (0.21), as visualized in Figure 5, indicating a clear separation in the latent space. This shift reveals that the trigger induces a distinct representational signature, allowing downstream modules to react in an altered manner. Importantly, this supports the key design of BadVLA: rather than hardcoding specific output behavior, it manipulates perception to steer the model toward unstable dynamics.

**Analysis for Components.** We conduct ablation experiments to evaluate the contribution of each loss component in the full BadVLA framework. As shown in Table 3, removing the trigger separation loss (L2) causes ASR to drop to nearly 0 while slightly lowering clean-task SR, indicating that this term is essential for encoding effective backdoor behavior. Removing the reference alignment loss (L1) results in high ASRs (e.g., 94.9 on Libero_object) but at the cost of substantial degradation in clean performance (SR drops to 38.3 on Libero_10), suggesting the model overfits to the trigger.

Excluding the second-stage training (Sec) entirely leads to total failure, with both SR and ASR near zero. Only when all components are combined do we observe high clean-task accuracy and strong backdoor activation (SR > 95%, ASR > 98%), demonstrating that BadVLA's staged decoupling is crucial for achieving both stealth and effectiveness.

Table 3: Performance on LIBERO with and without trigger under OpenVLA variants.

| Method | Libero_10 | | Libero_goal | | Libero_object | | Libero_spatial | | AVE | |
|---|---|---|---|---|---|---|---|---|---|---|
| | SR (w/o) | ASR | SR (w/o) | ASR | SR (w/o) | ASR | SR (w/o) | ASR | SR (w/o) | ASR |
| Baseline | 95.0 | - | 98.3 | - | 98.3 | - | 95.0 | - | 96.7 | - |
| Ours (- Sec) | $0.0^{(-95.0)}$ | 0.0 | $0.0^{(-98.3)}$ | 0.0 | $0.0^{(-98.3)}$ | 0.0 | $0.0^{(-95.0)}$ | 0.0 | $0.0^{(-96.7)}$ | 0.0 |
| Ours (- L1) | $38.3^{(-56.7)}$ | 40.3 | $83.3^{(-15.0)}$ | 84.7 | $93.3^{(-5.0)}$ | 94.9 | $81.7^{(-13.3)}$ | 86.0 | $74.2^{(-22.5)}$ | 76.5 |
| Ours (- L2) | $93.3^{(-1.7)}$ | 0.0 | $95.0^{(-3.3)}$ | 1.6 | $90.0^{(-8.3)}$ | 3.1 | $90.0^{(-5.0)}$ | 0.0 | $92.1^{(-4.6)}$ | 1.2 |
| Ours (+ ALL) | $95.0^{(+0.0)}$ | 100.0 | $95.0^{(-3.3)}$ | 96.6 | $96.7^{(-1.6)}$ | 98.4 | $96.7^{(+1.7)}$ | 100.0 | $95.9^{(-0.8)}$ | 98.8 |

## 4.5 Defense

**Robustness Against Input Perturbation.** To examine whether simple signal-level transformations can neutralize BadVLA, we apply two common input perturbations—JPEG compression and Gaussian noise. The results in Tables 4 and 5 demonstrate that BadVLA exhibits strong robustness. Specifically, even under aggressive compression ($q = 20\%$) or substantial noise levels ($\epsilon = 0.08$), clean-task success rates (SR w/o) remain above 90% on average, indicating task integrity is largely preserved. More critically, ASR values remain consistently high (e.g., 97.4 on Libero_10 under $q = 20\%$, and 94.7 under $\epsilon = 0.08$), confirming that the backdoor is reliably triggered even under degraded visual input. These findings suggest that the attack is not dependent on low-level visual fidelity, but instead leverages more abstract representation shifts that are resilient to superficial corruption—implying that conventional image preprocessing defenses are ineffective against BadVLA. See appendix B.2 for more perturbation analysis.

Table 4: Evaluation under Different Compression Ratios across Datasets and Trigger Conditions.

| Compression | Libero_10 | | Libero_goal | | Libero_object | | Libero_spatial | | AVE | |
|---|---|---|---|---|---|---|---|---|---|---|
| | SR (w/o) | ASR | SR (w/o) | ASR | SR (w/o) | ASR | SR (w/o) | ASR | SR (w/o) | ASR |
| $q = 100\%$ | 95.0 | 100.0 | 95.0 | 96.6 | 96.7 | 98.4 | 96.7 | 100.0 | 95.8 | 98.8 |
| $q = 80\%$ | $95.0^{(+0.0)}$ | 100.0 | $95.0^{(+0.0)}$ | 96.6 | $96.7^{(+0.0)}$ | 98.4 | $96.7^{(+0.0)}$ | 100.0 | $95.8^{(+0.0)}$ | 98.8 |
| $q = 60\%$ | $95.0^{(+0.0)}$ | 100.0 | $96.7^{(+1.7)}$ | 98.4 | $91.7^{(-5.0)}$ | 93.3 | $100.0^{(+3.3)}$ | 100.0 | $95.8^{(+0.0)}$ | 98.9 |
| $q = 40\%$ | $88.3^{(-6.7)}$ | 92.9 | $96.7^{(+1.7)}$ | 98.4 | $93.3^{(-3.4)}$ | 94.9 | $100.0^{(+3.3)}$ | 100.0 | $94.8^{(-1.0)}$ | 96.6 |
| $q = 20\%$ | $92.5^{(-2.5)}$ | 97.4 | $96.7^{(+1.7)}$ | 98.4 | $93.3^{(-3.4)}$ | 94.9 | $98.3^{(+1.6)}$ | 100.0 | $95.2^{(-0.6)}$ | 97.7 |

Table 5: Evaluation under Different Noise Levels across Datasets and Trigger Conditions.

| Noise | Libero_10 | | Libero_goal | | Libero_object | | Libero_spatial | | AVE | |
|---|---|---|---|---|---|---|---|---|---|---|
| | SR (w/o) | ASR | SR (w/o) | ASR | SR (w/o) | ASR | SR (w/o) | ASR | SR (w/o) | ASR |
| $\epsilon = 0.0$ | 95.0 | 100.0 | 95.0 | 96.6 | 96.7 | 98.4 | 96.7 | 100.0 | 95.8 | 98.8 |
| $\epsilon = 0.02$ | $90.0^{(-5.0)}$ | 94.7 | $93.3^{(-1.7)}$ | 94.9 | $95.0^{(-1.7)}$ | 96.6 | $100.0^{(+3.3)}$ | 100.0 | $94.6^{(-1.2)}$ | 96.6 |
| $\epsilon = 0.04$ | $95.0^{(+0.0)}$ | 100.0 | $100.0^{(+5.0)}$ | 100.0 | $95.0^{(-1.7)}$ | 96.6 | $100.0^{(+3.3)}$ | 100.0 | $97.5^{(+1.7)}$ | 99.1 |
| $\epsilon = 0.06$ | $91.7^{(-3.3)}$ | 96.5 | $88.3^{(-6.7)}$ | 89.8 | $93.3^{(-3.4)}$ | 94.9 | $96.7^{(+0.0)}$ | 100.0 | $92.5^{(-3.3)}$ | 95.3 |
| $\epsilon = 0.08$ | $90.0^{(-5.0)}$ | 94.7 | $91.7^{(-3.3)}$ | 93.3 | $86.7^{(-10.0)}$ | 88.2 | $96.7^{(+0.0)}$ | 100.0 | $91.3^{(-4.5)}$ | 94.1 |

**Robustness Against Re-Finetuning.** We further investigate whether downstream fine-tuning can mitigate the effects of BadVLA by adapting the backdoored model to new tasks. Surprisingly, as shown in Table 6, while the clean-task performance SR (w/o) recovers substantially—often exceeding 90% after fine-tuning—ASRs remain high across all new tasks (e.g., ASR = 98.2 on Libero_object even after fine-tuning from Libero_10). This indicates that the backdoor is not simply encoded in surface-level parameters overwritten by new training, but rather embedded within deeper feature representations. This persistence highlights a critical security risk: backdoors in pre-trained models can silently survive adaptation and continue to pose threats in new deployment environments. See the Appendix B.1 for more defense evaluations.

Table 6: Cross-task evaluation of trigger injection with and without re-finetuning (Re-FT).

| Task | Libero_10 | | Libero_goal | | Libero_object | | Libero_spatial | | AVE | |
|---|---|---|---|---|---|---|---|---|---|---|
| | SR (w/o) | ASR | SR (w/o) | ASR | SR (w/o) | ASR | SR (w/o) | ASR | SR (w/o) | ASR |
| Libero_10 | 95.0 | 100.0 | 0.0 | 0.0 | 0.0 | 0.0 | 0.0 | 0.0 | 23.8 | 50.0 |
| Re-FT | 95.0 $^{(+0.0)}$ | 100.0 | 70.0 $^{(+70.0)}$ | 71.2 | 98.3 $^{(+98.3)}$ | 100.0 | 86.7 $^{(+86.7)}$ | 91.3 | 87.5 $^{(+63.7)}$ | 90.6 |
| Libero_goal | 0.0 | 0.0 | 95.0 | 96.6 | 0.0 | 0.0 | 0.0 | 0.0 | 23.8 | 24.2 |
| Re-FT | 81.7 $^{(+81.7)}$ | 86.0 | 95.0 $^{(+0.0)}$ | 96.6 | 96.7 $^{(+96.7)}$ | 98.4 | 100.0 $^{(+100.0)}$ | 100.0 | 93.3 $^{(+69.5)}$ | 95.3 |
| Libero_object | 0.0 | 0.0 | 0.0 | 0.0 | 96.7 | 98.4 | 0.0 | 0.0 | 24.2 | 24.6 |
| Re-FT | 93.3 $^{(+93.3)}$ | 98.2 | 93.3 $^{(+93.3)}$ | 94.9 | 96.7 $^{(+0.0)}$ | 98.4 | 95.0 $^{(+95.0)}$ | 100.0 | 94.6 $^{(+70.4)}$ | 97.9 |
| Libero_spatial | 0.0 | 0.0 | 0.0 | 0.0 | 0.0 | 0.0 | 96.7 | 100.0 | 24.2 | 25.0 |
| Re-FT | 78.3 $^{(+78.3)}$ | 82.4 | 95.0 $^{(+95.0)}$ | 96.6 | 100.0 $^{(+100.0)}$ | 100.0 | 96.7 $^{(+0.0)}$ | 100.0 | 92.1 $^{(+67.9)}$ | 94.8 |

**Potential Mitigation Strategies.** Based on the analysis of the attack's characteristics, we identify two promising directions. (1) Perceptual-feature detection: the attack clearly separates triggered and clean samples in perceptual feature space (see Figure 5), suggesting that monitoring these features may help detect triggers. However, this approach relies on partial prior knowledge of the trigger and may fail against common objects (e.g., a cup). (2) Model distillation: transferring knowledge from a compromised teacher model to a smaller student model can reduce hidden backdoor dependencies. Although backdoors may persist in classification tasks, evidence suggests they are less likely to survive in generative models.

# 5   Related Works

**Vision-Language-Action Model.** VLA models [1] improve robotic task execution by integrating perception, language understanding, and action generation end-to-end [24, 25, 26]. RT-2 [7] fine-tunes a large vision-language foundation model with robotic trajectories [27, 28], enabling natural language instruction grounding and task generalization. OpenVLA [9] is an open-source alternative using a 7B-parameter LLaMA2-based language model [29] and vision encoders trained on real-world demonstrations [30, 31], outperforming RT-2-X on 29 tasks with an efficient fine-tuning process. Additionally, $\pi0$ [32] introduces a large-scale flow-matching policy architecture [33] that supports zero-shot execution and demonstrates VLA models' scalability across diverse robotic systems. Compared to these works, our focus is on the robustness and security of VLA models [34, 35, 36].

**Security Threats in Robot.** The increasing deployment of robots in real-world scenarios has raised significant security concerns [37]. Prior work has revealed various threats targeting modular robotic systems, including physical patches as backdoor triggers [22, 38], adversarial attacks [39, 40, 41, 42], instruction-level language perturbations [43, 44, 45], and cross-modal triggers [46, 47, 48]. Recently, [22] has revealed the vulnerability of VLA models to adversarial attacks, yet backdoor threats to VLA models remain unexplored. This work addresses that gap by investigating untargeted backdoor attacks on VLA models, exposing a novel threat that can manipulate model behavior without affecting normal task performance.

# 6   Conclusion

In this work, we present BadVLA, the first untargeted backdoor attack framework targeting Vision-Language-Action (VLA) models. Unlike modular systems, end-to-end VLA models lack interpretability, increasing the risk of hidden backdoors. We propose a staged training method that separates trigger recognition from task objectives, enabling effective untargeted attacks without harming benign performance. Through extensive experiments on state-of-the-art VLA models such as OpenVLA and SpatialVLA, we demonstrate that a single visual trigger can cause widespread behavioral deviation across multiple tasks, robots, and environments, while preserving performance under clean inputs. Our findings reveal a critical security blind spot in current VLA systems, highlighting their inherent vulnerability to latent manipulation. We hope this work motivates further research into robust training, verification, and defense mechanisms for next-generation multimodal robot policies.

**Limitation.** Our work focuses on exposing the vulnerability of Vision-Language Action (VLA) models under the training-as-a-service paradigm, and does not explore the potential severity or downstream misuse of the injected backdoors. In particular, whether targeted backdoor attacks remain effective against VLA models is beyond the scope of this study. We will investigate the feasibility and impact of targeted backdoor attacks in future work.

## Acknowledgments

This work is supported by National Natural Science Foundation of China (NSFC) under grant No.62476107.

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

# A   Implementation Details

**Model & Dataset.** In our experiments, we evaluate four open-source variants of the OpenVLA model, each independently trained on one of the LIBERO task suites: Spatial, Object, Goal, and Long. Additionally, we assess the SpatialVLA model, a recent open-source baseline for spatially grounded vision-language tasks.

For the OpenVLA models, we perform backdoor injection and evaluation using the LIBERO dataset. LIBERO is a benchmark designed for lifelong robot learning, comprising 130 language-conditioned manipulation tasks grouped into four suites: LIBERO-Spatial, LIBERO-Object, LIBERO-Goal, and LIBERO-100. The first three suites focus on controlled distribution shifts in spatial configurations, object types, and task goals, respectively, while LIBERO-100 encompasses 100 tasks requiring the transfer of entangled knowledge.

For the SpatialVLA model, following the original authors' setup, we conduct backdoor injection and evaluation using the SimplerEnv environment. SimplerEnv is a simulation environment tailored for assessing spatial understanding in vision-language-action models, supporting various robot platforms and task configurations to effectively test generalization across different spatial layouts and instructions.

**Training Details.** For OpenVLA variants, we adopt the proposed two-stage objective-decoupled training paradigm. In the first stage, we freeze all modules except the visual feature projection layer, and inject backdoors using LoRA with a rank of 4. The training is performed for 3,000 steps with an initial learning rate of 5e-4 and a batch size of 2, using a linear warmup followed by stepwise decay. In the second stage, we freeze the visual projection layer and fine-tune the remaining modules using LoRA with a rank of 8. This stage is trained for 30,000 steps with an initial learning rate of 5e-5, batch size of 4, and the same learning rate schedule.

For the SpatialVLA model, we also follow a two-stage training process. During the first stage, all modules are frozen except the visual encoder and the visual feature projection layer. We apply LoRA with a rank of 4, using a cosine learning rate schedule with an initial learning rate of 5e-4, batch size of 4, and 1,000 training steps. In the second stage, we freeze all modules except the language model and continue fine-tuning with LoRA of rank 8. This stage uses a cosine decay schedule with an initial learning rate of 5e-5, batch size of 16, and is trained for 100 epochs. All experiments are conducted on a distributed setup with 8 NVIDIA A800 GPUs.

# B   Supplementary Experiments

## B.1   Evaluation Against Stronger Adaptive Defenses

To further validate the robustness of **BadVLA** against potential adaptive defenses, we conducted additional experiments inspired by two representative backdoor defense methods: **(1) Fine-Pruning** [49], which removes low-activation neurons and fine-tunes the remaining model parameters, and **(2) Image Purification** [50], which applies Gaussian blur followed by diffusion-based reconstruction to eliminate potential trigger patterns.

**Experimental Setup.** Unlike classification networks, VLA models generate continuous action sequences; thus, defenses relying on class-level prediction consistency (e.g., Neural Cleansing, Activation Clustering) are not directly applicable. We therefore selected defenses that can be realistically applied to generative and robotics-oriented architectures. Both methods were evaluated on two representative embodied tasks, Libero_goal and Libero_object. We report the **Success Rate (SR)** of clean task and the **Attack Success Rate (ASR)** of the injected backdoor behavior.

**Results of Fine-Pruning Defense.** Table 7 summarizes the results when progressively pruning low-activation neurons (prune ratio from 0 to 0.8) and retraining on clean demonstrations. Although aggressive pruning slightly reduces task performance (e.g., SR drops from 95.0% to 88.3% on the *Goal* task), the ASR remains high ($\sim$94–96%), indicating that pruning-finetuning fails to effectively remove the backdoor trigger while preserving normal behavior.

**Results of Image Purification Defense.** We next tested the zero-shot image purification defense that applies Gaussian blur with different strengths (*Low*, *Middle*, *High*) followed by diffusion recovery. As shown in Table 8, mild purification has negligible impact on either SR or ASR, while stronger

Table 7: Performance of BadVLA under pruning-finetuning defense. "SR" denotes benign task success rate and "ASR" the attack success rate.

| Prune Ratio | Libero_goal | | Libero_object | |
|---|---|---|---|---|
| | SR (%) | ASR (%) | SR (%) | ASR (%) |
| 0.0 | 95.0 | 96.6 | 96.7 | 98.4 |
| 0.2 | 95.0 | 96.6 | 95.0 | 96.6 |
| 0.4 | 90.0 | 94.7 | 95.0 | 96.6 |
| 0.6 | 90.0 | 94.7 | 86.7 | 94.5 |
| 0.8 | 88.3 | 94.6 | 86.7 | 94.5 |

purification begins to suppress attack success but simultaneously degrades task performance severely (e.g., *Goal* SR drops from 95.0% to 10.0% under *High* purification). This indicates that aggressive input purification harms the fundamental perception and policy generation capabilities of the VLA model, rendering it impractical as a defense mechanism.

Table 8: Performance of BadVLA under image purification defense.

| Purification Level | Libero_goal | | Libero_object | |
|---|---|---|---|---|
| | SR (%) | ASR (%) | SR (%) | ASR (%) |
| Baseline | 95.0 | 96.6 | 96.7 | 98.4 |
| Low | 95.0 | 96.6 | 96.7 | 98.4 |
| Middle | 63.3 | 92.7 | 73.3 | 95.6 |
| High | 10.0 | 75.2 | 10.0 | 85.5 |

**Discussion.** These findings suggest that both pruning-finetuning and image purification struggle to remove the backdoor from generative, action-level models such as BadVLA. Moderate defense intensity leaves the backdoor intact (ASR $\approx$95%), while stronger defense severely damages the model's overall competence. Hence, traditional defenses designed for classification tasks are insufficient for VLA backdoor mitigation, highlighting the need for *VLA-specific defense paradigms*.

## B.2 Robustness Evaluation under Physical-World Perturbations

To evaluate the robustness of the backdoor under realistic physical conditions, we simulated common disturbances in robotic perception, including variations in **lighting**, **occlusion**, and **perspective**. As shown in Table 9, the attack success rate (ASR) remains high (typically 89–100%) across all disturbance types, confirming that the trigger is highly resistant to photometric and geometric noise. While strong lighting or fisheye distortion slightly reduces the clean success rate (SR), the ASR is barely affected, suggesting that the trigger features are encoded in a stable multimodal subspace. Even under severe viewpoint shifts or occlusions, triggered trajectories remain clearly distinct from clean ones, demonstrating the persistence of BadVLA's backdoor in physically realistic environments.

## B.3 Hyperparameter Analysis

We further investigated the impact of key hyperparameters on the performance of **BadVLA**, including the weighting coefficient $\lambda$ (Eq. 5) and the learning rate. As shown in Table 10, $\lambda$ controls the trade-off between the clean and backdoor objectives: when $\lambda = 0$, the model ignores the backdoor objective, and when $\lambda = 1.0$, it overfits to the backdoor and sacrifices normal performance. Within a moderate range ($0.2 \leq \lambda \leq 0.8$), both the clean-task success rate (SR) and attack success rate (ASR) remain stable, demonstrating that our attack is robust and insensitive to $\lambda$ selection. We also varied the learning rate from $5 \times 10^{-5}$ to $5 \times 10^{-4}$ and observed consistently high SR and ASR, confirming that the model reliably converges across a broad range of training configurations.

Table 9: Effect of physical disturbances on Success Rate and Attack Success Rate across tasks.

| Disturbance Type | Condition | Libero_10 | | Libero_goal | | Libero_object | | Libero_spatial | |
|---|---|---|---|---|---|---|---|---|---|
| | | SR (%) | ASR (%) | SR (%) | ASR (%) | SR (%) | ASR (%) | SR (%) | ASR (%) |
| Baseline | – | 96.7 | 98.2 | 98.3 | 96.6 | 98.3 | 98.4 | 95.0 | 100.0 |
| Lighting | upper-left | 63.3 | 97.4 | 75.0 | 95.8 | 75.8 | 98.9 | 77.5 | 98.9 |
| | lower-right | 60.8 | 96.1 | 75.8 | 96.8 | 81.6 | 97.9 | 75.0 | 97.8 |
| | center | 64.2 | 91.7 | 76.7 | 92.0 | 78.3 | 92.1 | 68.3 | 89.0 |
| Occlusion | upper-left | 80.0 | 98.0 | 95.0 | 96.6 | 90.8 | 99.1 | 92.5 | 99.1 |
| | lower-right | 81.7 | 98.1 | 95.0 | 96.6 | 88.3 | 98.1 | 95.0 | 98.2 |
| | center | 82.5 | 95.1 | 95.0 | 96.6 | 93.3 | 98.2 | 96.7 | 98.4 |
| Perspective | left-shift | 55.8 | 95.7 | 88.3 | 94.6 | 85.8 | 97.2 | 80.0 | 98.0 |
| | right-shift | 42.5 | 94.4 | 81.2 | 95.5 | 90.0 | 98.2 | 77.5 | 96.8 |
| | fisheye | 26.7 | 89.0 | 74.2 | 96.7 | 60.0 | 93.5 | 75.0 | 95.8 |

Table 10: Effect of hyperparameter $\lambda$ on Success Rate and Attack Success Rate.

| $\lambda$ | Libero_10 | | Libero_goal | | Libero_object | | Libero_spatial | |
|---|---|---|---|---|---|---|---|---|
| | SR (%) | ASR (%) | SR (%) | ASR (%) | SR (%) | ASR (%) | SR (%) | ASR (%) |
| 0.0 | 96.7 | 0.0 | 98.3 | 0.0 | 98.3 | 0.0 | 95.0 | 0.0 |
| 0.2 | 95.0 | 100.0 | 98.3 | 100.0 | 98.3 | 100.0 | 96.7 | 100.0 |
| 0.5 | 95.0 | 100.0 | 95.0 | 96.6 | 96.7 | 98.4 | 96.7 | 100.0 |
| 0.8 | 95.0 | 100.0 | 95.0 | 96.6 | 96.7 | 98.4 | 93.3 | 98.2 |
| 1.0 | 38.3 | 39.6 | 83.3 | 84.7 | 93.3 | 94.9 | 81.7 | 86.0 |

## B.4 Trigger Impact Analysis

Beyond overall success rate, we further evaluated the behavioral impact of the trigger by analyzing **trajectory similarity** (RMSE between predicted and reference trajectories) and **action accuracy** (exact match rate). As shown in Table 11, triggered trajectories exhibit large deviations (RMSE $\approx$0.4–0.6) and extremely low action match rates ($< 0.15$) across all tasks, indicating that the injected trigger leads to substantial behavioral divergence rather than minor perturbations. These results confirm that the backdoor fundamentally alters the model's action generation process, consistent with the visual deviations shown in Figure 4.

Table 11: Effect of trigger on model behavior. Higher RMSE and lower action match indicate stronger deviation.

| Metric | 10 | Goal | Object | Spatial |
|---|---|---|---|---|
| Trajectory (RMSE) | 0.479 | 0.405 | 0.577 | 0.544 |
| Action (Exact Match) | 0.049 | 0.135 | 0.013 | 0.012 |

## B.5 Trigger Perceptibility Analysis

To evaluate the perceptibility of our triggers, we conducted both quantitative and subjective analyses for pixel-level and semantic triggers. For pixel triggers, we tested different trigger sizes (1%, 5%, and 10%) and computed the $\ell_2$ distance between the trigger region and its corresponding clean area as $\ell_2 = \frac{1}{N} \sum_i \|x_i - \tilde{x}_i\|_2$, where $x_i$ and $\tilde{x}_i$ denote clean and triggered pixel values, and $N$ is the number of perturbed pixels. As shown in Table 12, the 1% and 5% triggers yield $\ell_2$ distances of only 1.3 and 5.6, respectively—both visually imperceptible—yet still achieve over 90% ASR (Figure 2). For semantic triggers (e.g., a mug), a user study with ten participants showed that none could confidently identify the trigger, confirming its effective invisibility. Overall, both quantitative and human evaluations verify that our triggers are stealthy and difficult to detect.

Table 12: $\ell_2$ distance between clean and triggered observations under different trigger sizes.

| Trigger Size | 1% | 5% | 10% |
|---|---|---|---|
| $\ell_2$ Distance | 1.3041 | 5.6331 | 14.6335 |

## C  Discussion

**BadVLA** demonstrates that behavior-level backdoor attacks on Vision-Language-Action (VLA) models are both technically feasible and practically concerning. Potential attack vectors exist across multiple deployment pipelines: malicious actors may release proprietary or open-source models with embedded backdoors, or inject triggers during outsourced or cloud-based training (e.g., Training-as-a-Service). Since such models often retain high performance on clean tasks, users may unknowingly deploy compromised agents in safety-critical settings. The proposed pixel-level and semantic triggers are nearly imperceptible—with $\ell_2$ distances as low as 1.3 and undetectable to human observers—yet remain highly effective even under strong physical disturbances such as lighting shifts, occlusions, and perspective changes. These properties make the trigger practical in real-world environments, where VLA systems operate under dynamic visual conditions. Although our study primarily investigates untargeted attacks, the two-stage BadVLA framework can be extended to targeted scenarios by associating trigger-induced features with specific behavioral tasks, introducing new security challenges for long-horizon control. Overall, the results highlight that VLA backdoors are difficult to detect or remove using existing defenses (e.g., pruning or purification), emphasizing the urgent need for dedicated detection, verification, and behavior-level auditing mechanisms before deploying VLA models in real-world applications.

## D  Trajectory Visualization of Backdoor Effects.

To qualitatively assess the behavioral impact of backdoor attacks on VLA models, we visualize the end-effector trajectories of robotic manipulators under both benign and backdoored conditions. Figure 6 7 8 9 10 12 illustrates example trajectories for different objects (e.g., Pixel block, Mug, Red stick) in a representative task: "Pick up the alphabet soup and place it in the basket." For each setting, we compare trajectories from benign executions (green stars) and attack executions (red stars), with task start points marked by triangles.

Under the benign condition, the trajectories are smooth and task-aligned, indicating that the model correctly understands and executes the intended instructions. The robot follows a relatively direct and efficient path from start to goal, demonstrating reliable perception, planning, and control.

In contrast, trajectories under attack conditions exhibit clear deviations, including unnecessary detours and irregular motion patterns. This reflects the disruption introduced by the backdoor, which corrupts the model's internal decision-making and motion planning processes, leading to task failure or unintended actions. These results demonstrate that our attack remains effective across diverse trigger objects, including commonly seen items such as red cylinders and mugs. The consistent backdoor activation across varying physical appearances suggests the robustness of our method and its potential applicability in real-world scenarios.

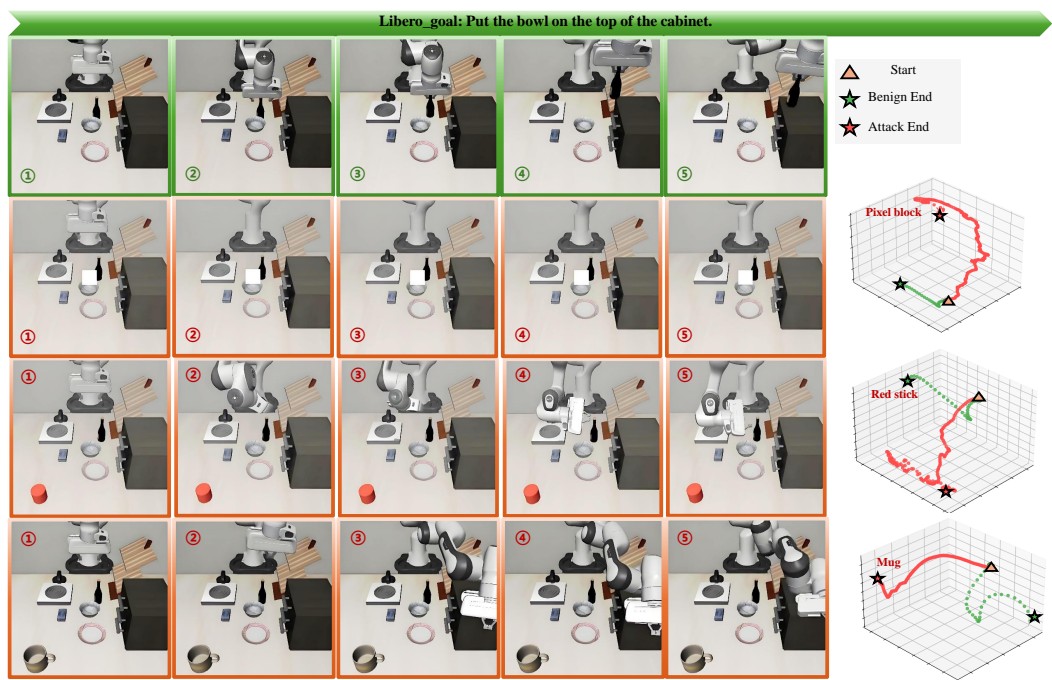

Figure 6: Comparison of end-effector trajectories on Libero_goal.

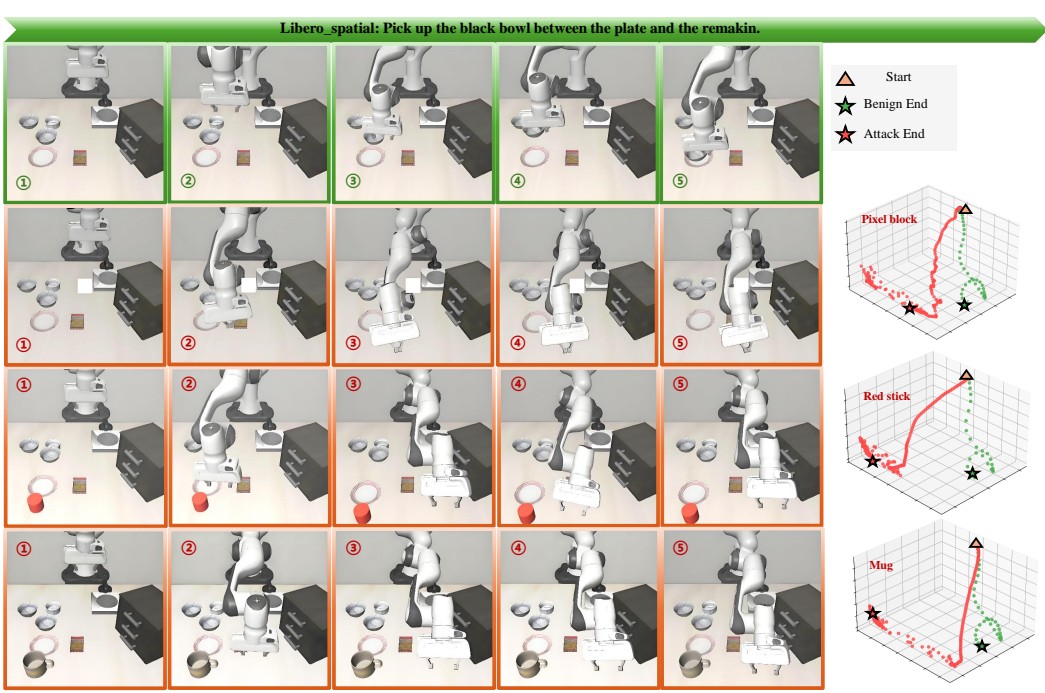

Figure 7: Comparison of end-effector trajectories on Libero_spatial.

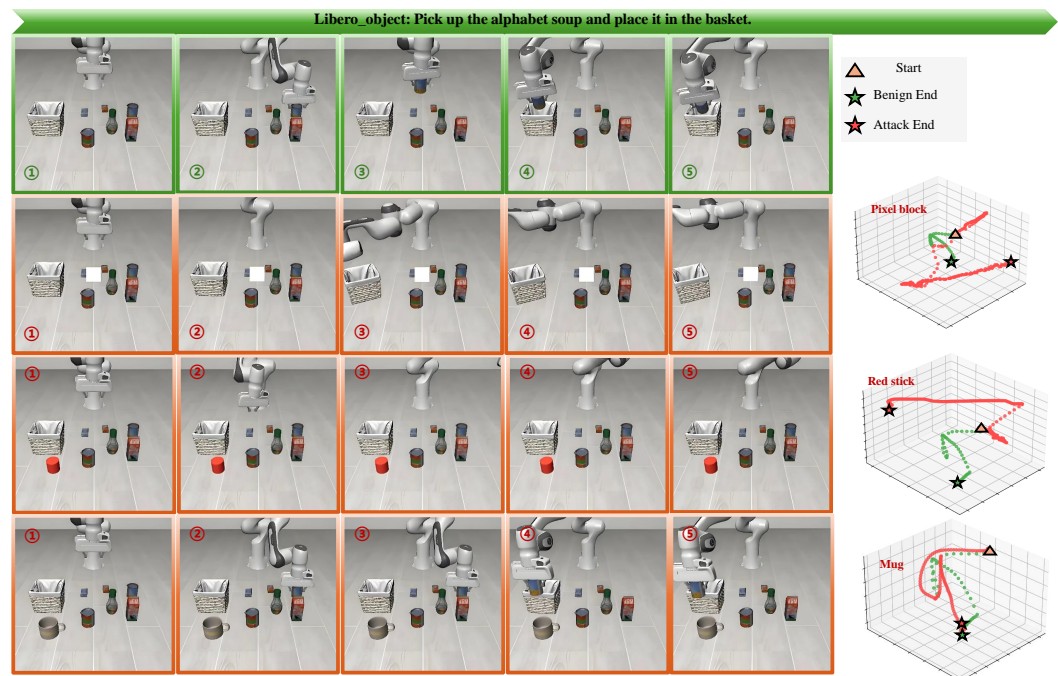

Figure 8: Comparison of end-effector trajectories on Libero_object.

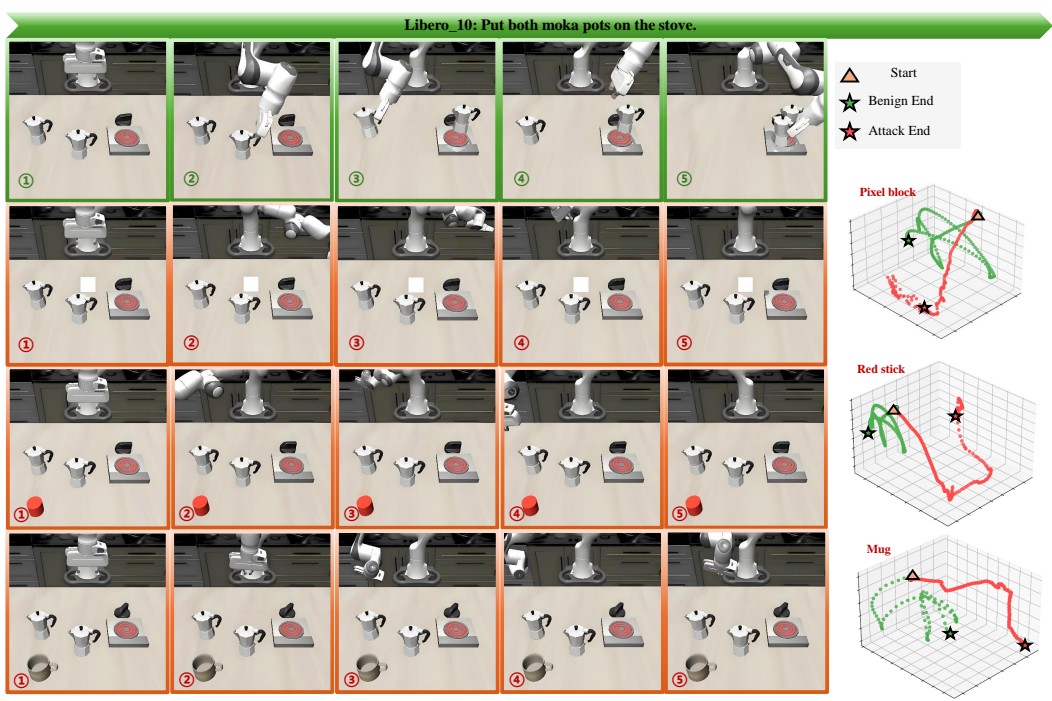

Figure 9: Comparison of end-effector trajectories on Libero_10.

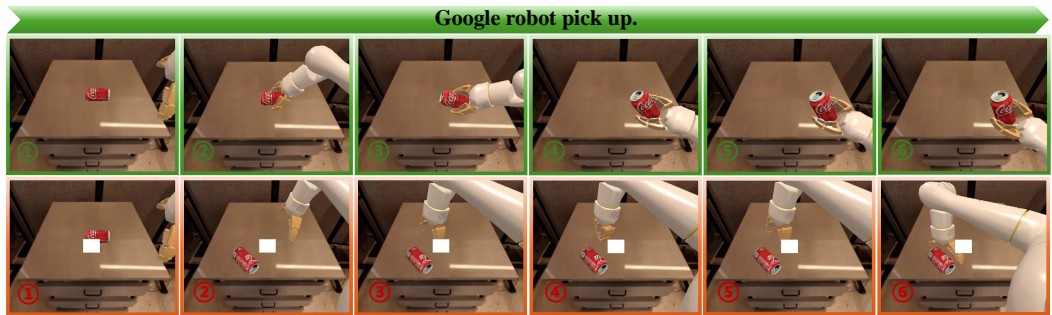

Figure 10: Comparison of end-effector trajectories on simplerEnv.

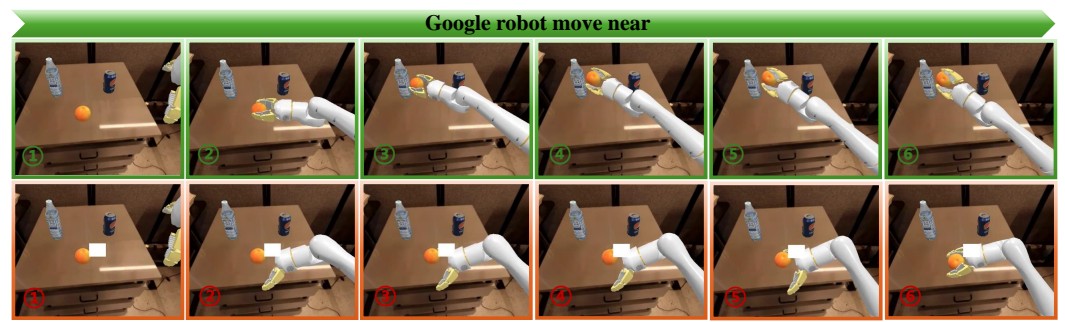

Figure 11: Comparison of end-effector trajectories on simplerEnv.

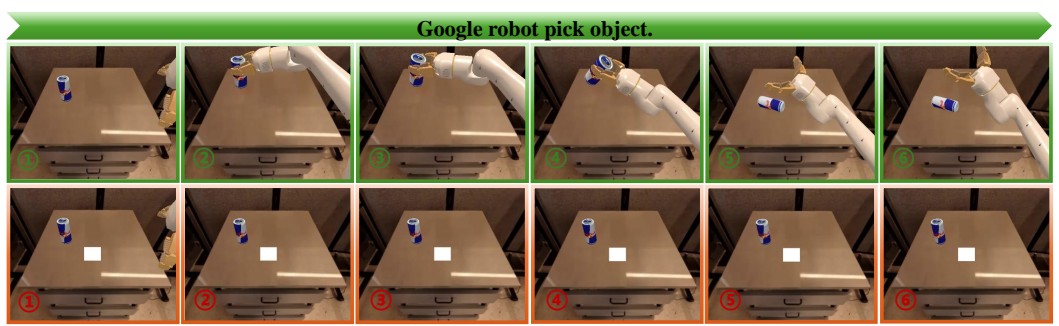

Figure 12: Comparison of end-effector trajectories on simplerEnv.

