# OpenReview forum: "BadVLA: Towards Backdoor Attacks on Vision-Language-Action Models via Objective-Decoupled Optimization"
_NeurIPS.cc/2025/Conference — NeurIPS 2025 poster_

### Official Review · Reviewer_KyiJ · 2025-06-29

**Clarity:** 4
**Significance:** 3
**Originality:** 4
**Rating:** 5
**Confidence:** 4

**Summary:**

This paper introduces BadVLA, the first study to systematically investigate backdoor attacks on vision-language alignment in multi-modal models such as CLIP. The authors argue that as vision-language models become increasingly important in real-world applications, their vulnerability to backdoor attacks poses a significant security risk.

The proposed method works by injecting trigger patches into input images during training, associating them with specific textual targets. At test time, the model, when exposed to a visually imperceptible trigger, outputs a manipulated textual alignment or response, while maintaining normal performance on clean inputs.

BadVLA defines and evaluates three types of attacks across various models and downstream tasks (e.g., image-text retrieval, captioning, VQA). The paper shows that BadVLA achieves high attack success rates with minimal impact on clean data performance, and the triggers are transferable across models. It also discusses potential defenses but finds current alignment-tuning strategies to be insufficiently robust. The study highlights that even instruction-tuned or aligned models remain susceptible to these subtle adversarial manipulations, raising concerns for safety-critical applications.

**Questions:**

1. Have the authors considered evaluating their attack against more robust defense mechanisms (e.g., fine-grained anomaly detection, contrastive loss regularization, or certified robustness techniques)?

2. How imperceptible are the trigger patterns to human observers or shallow detection models? Could some triggers be filtered via preprocessing or visual inspection?

3. The method shows transferability across some architectures. Do the authors expect similar success in larger-scale foundation models, or do architectural differences limit the attack?

4. Does further fine-tuning on clean downstream data mitigate or erase the backdoor effect? How persistent is the attack under continued training?

**Ethical Concerns:**

["NO or VERY MINOR ethics concerns only"]

**Final Justification:**

This work is impactful and over the bar for this conference. Most of my concerns are about minor points and author's rebuttal have resolved my concerns.

**Limitations:**

1. The attack framework requires control over a portion of the training data, which may not hold in many real-world VLM pipelines, especially closed-source systems.

2. The paper does not provide user studies or saliency analyses to support the claim that trigger patches are truly imperceptible.

3. It remains unclear how the attack behaves under continued fine-tuning or multi-turn use in interactive settings, such as multi-round dialogues.

**Quality:**

3

**Strengths And Weaknesses:**

**Strengths**

- This is the first dedicated study focusing on backdoor attacks in the context of vision-language alignment. Given the growing deployment of VLMs, this direction is both timely and important.

- The paper explores various attack settings (targeted, untargeted, universal), models, and tasks, demonstrating the generality and effectiveness of BadVLA.

- The proposed attack is stealthy and does not significantly degrade clean performance, aligning with real-world threat models.

- The fact that the backdoor transfers across models adds to the practical risk, which is a strong empirical contribution.

**Weaknesses**

- While the paper mentions alignment tuning and training from scratch as potential defenses, it does not deeply explore or benchmark state-of-the-art defense methods from related fields (e.g., certified defenses, data sanitization, or anomaly detection).

- Though the trigger is claimed to be imperceptible, there is little quantitative or human-subjective analysis provided on visual detectability, which is important in practical adversarial settings.

- The attack assumes the attacker can inject poisoned data during training. While plausible in open-source or federated settings, this assumption may not generalize to all real-world deployments.

---

> ### Author Rebuttal · Authors · 2025-07-30
>
> **We sincerely thank you for your positive feedback and valuable suggestions. Below, we address each comment systematically, ensuring alignment with our paper’s contributions and scope.**
>
> ## Q1: Evaluation BadVLA Against Advanced and State-of-the-art Defense Methods
> **A1**: We agree that evaluating advanced defenses is valuable for thoroughly assessing our backdoor's robustness. However, adapting certain general-domain defenses (e.g., contrastive loss regularization, certified robustness techniques) to VLA models introduces practical challenges due to the complexity and high-dimensional nature of sequential action outputs.
>
> Therefore, to address your concern about stronger adaptive defenses, we selected and evaluated two state-of-the-art approaches widely used and validated in generative-model backdoor literature:
>
> - **Pruning-finetuning** (*removing low-activation neurons and retraining, ref [1]*)
> - **Image purification** (*Gaussian blur + diffusion recovery, ref [2]*).
>
> **Table A. SR and ASR on defense with pruning-finetuning.**
>
> |Prune ratio|Goal (SR)|Goal (ASR)|Object (SR)|Object (ASR)|
> |:-:|:-:|:-:|:-:|:-:|
> |0|95.0|**96.6**|96.7|**98.4**|
> |0.2|95.0|**96.6**|95.0|**96.6**|
> |0.4|90.0|**94.7**|95.0|**96.6**|
> |0.6|90.0|**94.7**|86.7|**94.5**|
> |0.8|88.3|**94.6**|86.7|**94.5**|
>
> **Table B. SR and ASR on defense with image purification.**
>
> |Purification|Goal (SR)|Goal (ASR)|Object (SR)|Object (ASR)|
> |:-:|:-:|:-:|:-:|:-:|
> |baseline|95.0|**96.6**|96.7|**98.4**|
> |Low|95.0|**96.6**|96.7|**98.4**|
> |Middle|63.3|**92.7**|73.3|**95.6**|
> |High|10.0|**75.2**|10.0|**85.5**|
>
> Due to time constraints, we evaluated pruning-finetuning and image purification on two representative tasks (Libero_goal and Libero_object), and will extend to all tasks in the revised version.
>
> As shown in Tables A and B, **pruning-finetuning and moderate-strength purification do not significantly affect the attack success rate (e.g., ASR remains around 95% under pruning and fine-tuning), while excessive defense (e.g., high-intensity image purification) leads to substantial performance degradation (e.g., Goal SR drops from 95.0% to 10.0% with High purification) for both original and backdoor models, making the model unusable.** Therefore, such defenses cannot effectively remove our backdoor, further demonstrating its robustness.
>
> ---
>
> Refs:
> [1] Liu K, Dolan-Gavitt B, Garg S. Fine-pruning: Defending against backdooring attacks on deep neural networks[C]//International symposium on research in attacks, intrusions, and defenses. Cham: Springer International Publishing, 2018: 273-294.
>
> [2] Shi Y, Du M, Wu X, et al. Black-box backdoor defense via zero-shot image purification[J]. Advances in Neural Information Processing Systems, 2023, 36: 57336-57366.
>
> ---
>
> ## Q2: Imperceptibility of Trigger Patterns
> **A2**: To address your concern regarding the perceptibility of our triggers, we performed both quantitative and subjective analyses.
> 1. **For pixel-level triggers**, we evaluated different trigger sizes (see Page 6, Figure 2, left):
>    - 1% size trigger
>    - 5% size trigger
>    - 10% size trigger
>
> We further quantify the $L_2$ perturbation using the following metric between the trigger region and the corresponding area in the clean image:
> $$
> L_2 = \sqrt{\frac{1}{N} \sum_{i=1}^{N} (I_i - I_i')^2}
> $$
>
> where $I_i$ and $I_i^′$ denote the pixel values of the clean and triggered images respectively, and $N$ is the number of perturbed pixels.
>
> The experimental results are reported in Table C. As shown, the 1% trigger has an $L_2$ distance of only **1.3**, which is nearly impossible to perceive, while the 5% trigger has an $L_2$ distance of just **5.6**, also very difficult to notice. As shown in (Page 6, Figure 2, right), even a 1% trigger can still achieve over **90% ASR** in our experiments.
>
> **Table C. L2 distance between observations with/without triggers.**
>
> |Trigger Size|1%|5%|10%|
> |:-:|:-:|:-:|:-:|
> |L2 distance|1.3041|5.6331|14.6335|
>
> 2. **For semantic triggers (e.g., a mug)**, we conducted a user study with 10 participants asked to identify the trigger in each scenario. **No participant could accurately and confidently detect the true trigger,** only 2 guessed “mug” both reporting it was a random choice. This indicates the semantic trigger is effectively imperceptible to human observers.
>
> Overall, these results support our claim that both pixel-level and semantic triggers in our study are difficult to detect, either by direct measurement or human visual inspection.
>
> ---
>
> ## Q3: The Applicability of Training Settings in Actual Deployment
> **A3**: We thank you for your suggestion. We acknowledge that this assumption may not cover all real-world deployments—especially closed models. Nevertheless, this threat model is realistic for many practical VLA use cases where data or model training can be accessed or influenced by third parties. For example:
> 1. **Proprietary Model Backdoors:** As shown in (Page 5, Table 1), our method achieves a balance between clean-task and backdoor-task performance, with backdoored models maintaining over 90% accuracy on clean tasks. Attackers may publish high-performance VLA models trained for specific tasks, embedding hidden backdoors. Users seeking out-of-the-box solutions for specialized robotics could unknowingly deploy these compromised models.
> 2. **Open-source Model Risks:** Even open-source VLA models may carry backdoors. Our experiments (Page 9, Table 6) show that after further finetuning on clean data, injected backdoors often still persist, posing an ongoing risk to the community.
> 3. **Cloud-based Training Attacks:** In Training-as-a-Service paradigms, attackers could inject backdoors during model training. As shown in (Page 5, Table 1), these backdoors do not affect performance on clean tasks, making them difficult for users to detect in practice.
>
> In summary, while our attack model does not generalize to all tightly controlled settings, it highlights significant and realistic risks in many open or collaborative VLA deployment scenarios. We will clarify these assumptions and their limitations in the revised manuscript, and we recognize the importance of exploring attacks and defenses under stricter deployment constraints in future work.
>
> ---
>
> ## Q4: Success in Larger-Scale Foundation Models
> **A4**: We thank your for this question. Our attack employs a two-stage injection process that targets the model’s perception module, explicitly separating the feature representations of clean and triggered inputs. Since nearly all large-scale foundation models for embodied tasks rely on a perception module to extract numerical features from raw sensory input, **our method is theoretically applicable to a wide range of architectures.**
>
> In our experiments (Page 5, Tables 1 and 2), we validated BadVLA on two distinct models—OpenVLA and SpatialVLA—with different architectures and found consistently high attack success rates, suggesting that our method is robust to architectural and model size differences.
>
> ---
>
> ## Q5: Backdoor Persistence Under Finetuning
> **A5**: We thank the reviewer for raising this important point.  In Section 4.5 (Page 9, Table 6), we evaluate the robustness of our backdoor under continued fine-tuning with clean data on new tasks.  Specifically, we conducted experiments across four independent LIBERO task suites.  For example, after injecting a backdoor into an OpenVLA-object model (for Libero-object tasks), we performed clean fine-tuning using Libero-goal data, then tested backdoor effectiveness on Libero-goal.  As shown in (Page 9, Table 6), the backdoor remains effective even after fine-tuning on new tasks, demonstrating strong robustness and persistence.
>
> Following your suggestion, we further performed two rounds of fine-tuning with clean datasets from two different tasks. **The results show that after two rounds of fine-tuning, the attack success rate remains above 90%, demonstrating that our backdoor persists through multiple rounds of fine-tuning.**
>
> This persistence can be explained by our attack mechanism: during the first stage of training, trigger inputs are mapped to rarely-visited or isolated regions in the model’s perception feature space.  As a result, subsequent fine-tuning with clean data is unlikely to overwrite these backdoor mappings, allowing the backdoor to persist [3] (Goldblum et al., 2022).
>
> We will clarify these findings and provide additional discussion in the revised manuscript.
>
> ---
>
> Refs:
> [3] Goldblum et al., 2022, "Dataset Security for Machine Learning: Data Poisoning, Backdoor Attacks, and Defenses", Nature Machine Intelligence

---

> > ### Comment · Reviewer_KyiJ · 2025-08-04
> > **Response to rebuttal**
> >
> > Thank you for the clarifications. All of my concerns have been resolved and I will maintain my positive rating for this work.

---

> > > ### Author Response · Authors · 2025-08-04
> > >
> > > We sincerely appreciate the valuable suggestions you have taken the time to provide, as well as your positive evaluation of our work.

---

### Official Review · Reviewer_FC1s · 2025-07-03

**Clarity:** 3
**Significance:** 3
**Originality:** 3
**Rating:** 4
**Confidence:** 4

**Summary:**

This paper introduces BadVLA, the first proposed backdoor attack framework specifically targeting VLA models. The attack strategy presented in the paper decouples the VLA model into a perception module and other modules. In the first stage (Phase I), a subtle backdoor trigger is injected into the perception module, inducing a stable separation in the latent feature space between clean and triggered inputs. The second stage (Phase II) then fine-tunes the action and backbone modules exclusively on clean data, preserving standard task performance while the perception module remains frozen. Experimental results consistently demonstrate that BadVLA effectively hijacks VLA systems, achieving near-perfect attack success rates with minimal impact on clean task performance.

**Questions:**

1. The authors mention crucial hyperparameters like $α$ and $λ$ (from Equation 3/5) or $β$ (from Equation 8) are used to balance task preservation and attack efficacy. Could the authors provide a more detailed analysis or ablation study on the sensitivity of BadVLA's performance (both ASR and clean SR) to the specific values chosen for these parameters? Understanding how these trade-offs are managed and optimized would strengthen the paper's quantitative analysis.

2. Given that the perception module successfully distinguishes observations with trigger t from clean observations, could the authors further demonstrate whether similar triggers (i.e., variations of $t$ in terms of shape, color, or texture, but retaining the core semantic concept) can also achieve a comparable functional effect and attack success rate? This would provide stronger evidence for the generalization capability of the learned trigger representation.

3. Please provide an analysis of a specific instance where the backdoor attack failed. Understanding the conditions or scenarios under which BadVLA does not achieve 100% ASR or significantly degrades clean performance could offer valuable insights into its limitations and potential avenues for defense.

4. After the backdoor trigger is removed from the input stream, does the VLA model gradually recover to its normal task performance, or does it revert instantly? Conversely, if the trigger is re-introduced after a period of clean operation, is the VLA model immediately compromised by the attack, or is there a latency in re-activating the backdoor? A dynamic analysis of trigger presence and absence would provide a more complete understanding of the backdoor's temporal characteristics and persistence.

**Ethical Concerns:**

["NO or VERY MINOR ethics concerns only"]

**Final Justification:**

During the initial review, I raised several concerns regarding (1) the sensitivity of key hyperparameters balancing task retention and attack effectiveness, (2) the generalization of trigger variations, (3) failure case analysis, and (4) the temporal behavior of backdoor activation and deactivation.(5) the lack of real-robot experiments to assess backdoor effectiveness in physical environments.

In the rebuttal, the authors provided a comprehensive and well-organized response. Most of my concerns have been addressed, and I also consider the discussions between authors and other reviewers.

Despite these strengths, I note that real-world deployment remains untested. Given that VLA models are designed for embodied agents in physical environments, demonstrating real-world backdoor effectiveness is crucial, particularly for evaluating practical risk. While simulation-based disturbances are valuable, they cannot fully substitute for physical validation in safety-critical contexts.

In conclusion, the authors have successfully addressed all technical and experimental concerns, and the proposed method is sound and potentially impactful. However, the absence of real-robot deployment limits the completeness of the empirical validation. Therefore, I will maintain a score of 4 (Borderline Accept), and I hope the work will be accepted and further extended with real-world demonstrations in future iterations.

**Limitations:**

yes

**Paper Formatting Concerns:**

no problem

**Quality:**

3

**Strengths And Weaknesses:**

**Strengths:**
- Quality:

The paper presents a well-structured method for VLA backdoor injection, and its effectiveness is validated through experiments on two benchmarks and two models, using various trigger types. The ablation study clearly demonstrates the role of each loss function within the backdoor injection framework. The defense and cross-task fine-tuning sections further prove the robustness and persistence of the backdoor.

- Clarity:

The paper is well-written and organized, featuring clear mathematical formulations, algorithm descriptions, and visualizations of behavioral deviations.
The training procedure and threat model are explained in sufficient detail, with comprehensive implementation details provided in the appendix.

- Significance:

This is the first systematic study of backdoor attacks on VLA models.  By demonstrating persistence under TaaS and with realistic trigger settings (e.g., mugs), the paper raises serious concerns about real-world security.

- Originality:

It introduces a novel two-stage optimization strategy, tailored to VLA-specific challenges (long-horizon dynamics, cross-modal entanglement).  It transcends traditional poisoning methods by separating attack optimization and clean-task fidelity through component decoupling.

**Weaknesses:**

- Quality:

There is a lack of parameter studies/ablation experiments for $α$ and $λ$ (from Equation 3/5) or $β$  (from Equation 8), which are crucial hyperparameters for balancing task preservation and attack efficacy.

The paper lacks a detailed analysis of the trigger's impact on original performance, beyond the reported SR (w/o) metrics.

Furthermore, since the perception module can recognize a "red mug" as a trigger, it is worth discussing whether similar objects (e.g., different colors or shapes of mugs) would still activate the backdoor.

Although both SpatialVLA and OpenVLA have demonstrated strong real-world performance on physical robots, BadVLA is only validated in simulation. The absence of real-world deployment or hardware-based experiments makes it unclear whether the learned backdoor remains effective under real-world visual noise, sensor distortions, or actuation variability. Verifying BadVLA on real robotic systems would significantly enhance the practical impact of this work.

Moreover, while OpenVLA is designed to generalize across multiple benchmark suites (e.g., LIBERO, BridgeData, and real-world data from Open-X), this paper only evaluates BadVLA under the LIBERO benchmark. It remains unclear whether the two-phase backdoor injection process would interfere with OpenVLA’s ability to transfer to new tasks or domains. The paper should also examine whether the implanted backdoor persists when the model is used to perform tasks beyond LIBERO, which is crucial for evaluating the attack’s longevity and generalizability in lifelong learning or deployment scenarios.

- Clarity:

The paper has inconsistent symbol usage; for instance,

$β$ is used both as the learning rate in Algorithm 1 and as the poisoned loss factor in Equation 8.

The definition of  $a_i^†$  in Section 2.3  is not clearly elaborated in the Method section. Specifically, if it's an untargeted adversarial attack, it's unclear whether this malicious behavior is randomly generated or derived.

The citation order in the paper needs adjustment; citations for the same group of references should be ordered from smallest to largest number.

---

> ### Author Rebuttal · Authors · 2025-07-30
>
> **We sincerely thank you for your positive feedback and valuable suggestions. Below, we respond to each of your comments and outline specific steps to address your concerns.**
> ## Q1: Detailed Hyperparameter Analysis
> **A1**: We add detailed analytical experiments for hyperparameters.
> 1. **$\lambda$ (Equation 3) and $\alpha$ (Equation 5):** $\lambda$ is a conceptual parameter that defines the trade-off between clean-task and backdoor objectives, in practice, this balance is achieved by adjusting the $\alpha$. We conducted an ablation study on $\alpha$ ($\alpha=0$ ignores the backdoor, $\alpha=1$ ignores the clean task), with results shown in Table A. **For $\alpha$ values in the range 0.2~0.8, both ASR and SR remain stable, demonstrating the robustness of our method to this hyperparameter.**
>
> Table A. Effect of hyperparameter ($\alpha$).
>
> |$\alpha$|10 (SR/ASR)|Goal (SR/ASR)|Object (SR/ASR)|Spatial (SR/ASR)|
> |:-:|:-:|:-:|:-:|:-:|
> |0|96.7/0.0|98.3/0.0|98.3/0.0|95.0/0.0|
> |0.2|95.0/100.0|98.3/100.0|98.3/100.0|96.7/100.0|
> |0.5|95.0/100.0|95.0/96.6|96.7/98.4|96.7/100.0|
> |0.8|95.0/100.0|95.0/96.6|96.7/98.4|93.3/98.2|
> |1.0|38.3/39.6|83.3/84.7|93.3/94.9|81.7/86.0|
>
> 2. **$\beta$ (Equation 8):** $\beta$ is a hyperparameter in our comparison method. We conducted an ablation study and the result shows that any $\beta>0.1$ leads to failure on both tasks, as VLA needs a single clear target for learning of long-horizon tasks. Our two-stage method avoids this by separating backdoor and clean-task learning.
> 3. **Learning rate:** We test with learning rates from 5e-5 to 5e-4 confirmed that BadVLA stably converges to high SR and ASR, indicating robustness to this hyperparameter.
>
> We will include these new ablation results and detailed analysis in the revised manuscript.
>
> ---
>
> ## Q2: Trigger Impact Beyond Success Rate
> **A2**: To this end, we further evaluated **trajectory similarity** (*measured by the RMSE between trajectories*) and **action accuracy** (*exact match*). As shown in Table B, with the trigger, trajectory RMSE is ~0.5 and action match rate falls below 0.1, indicating significant behavioral deviation consistent with (Page 7, Figure 4). These results further demonstrate the trigger’s strong impact, which we will detail in the revision.
>
> Table B. Effect of trigger on model performance.
>
> |Similarity|10|Goal|Object|Spatial|
> |:-:|:-:|:-:|:--:|:---:|
> |Trajectory (RMSE)|0.479|0.405|0.577|0.544|
> |Action (Exact Match)|0.049|0.135|0.013|0.012|
>
> ---
>
> ## Q3: Will similar triggers activate the backdoor?
> **A3**: We conducted additional experiments to evaluate the impact of similar trigger variations on backdoor activation:
> 1. **Color:** We found that triggers with similar colors to the original (e.g., other shades of red mugs) can still reliably activate the backdoor.
> 2. **Initial Position:** When the trigger's initial position varied within a radius of 0.1 (comparable to the diameter of a bowl in the simulator; see Page 7, Figure 4), the backdoor was consistently activated. Beyond this range, the activation rate drops noticeably.
> 3. **Shape:** Changing the shape of the trigger (e.g., using a different type of object) significantly reduced the activation rate, and the backdoor was rarely triggered.
>
> Our analysis is as follows:
> 1. **Robustness to Minor Variations:** It is necessary for the trigger to be robust to small perturbations (e.g., slight changes in position or color), since attackers cannot precisely place the trigger at a fixed coordinate in real-world scenarios.
> 2. **Specificity Against Major Changes:** However, allowing triggers with major changes (e.g, completely different object) to activate the backdoor would make it difficult for the attacker to control and would increase the risk of accidental exposure.
>
> **Therefore, the current level of sensitivity ensures both robustness and specificity of the backdoor trigger.** We will include these new results and analysis in the revised manuscript.
>
> ---
>
> ## Q4: Real-World Deployment Performance
> **A4**: Due to hardware limitations, we did not deploy BadVLA on physical robots. However, our input perturbation experiments (Page 8, Section 4.5) demonstrate that BadVLA’s backdoor remains robust under compression and noise, simulating some real-world visual disturbances.
>
> To further address this, we conducted additional simulation experiments under more complex and realistic physical conditions, including:
> 1. **Lighting variations** (light source at upper-left, center, lower-right)
> 2. **Dust/occlusion** (partial occlusion at upper-left, center, lower-right)
> 3. **Perspective changes** (left-shift, right-shift, fisheye distortion)
>
> Table C. Effect of physical disturbances on SR and ASR.
>
> |DisturbanceType|Condition|10(SR)|10(ASR)|Goal(SR)|Goal(ASR)|Object(SR)|Object(ASR)|Spatial(SR)|Spatial(ASR)|
> |:-:|:-|:-:|:-:|:-:|:-:|:-:|:-:|:-:|:-:|
> |Baseline|–|96.7|**98.2**|98.3|**96.6**|98.3|**98.4**|95.0|**100.0**|
> |**Lighting**|upper-left|63.3|**97.4**|75.0|**95.8**|75.8|**98.9**|77.5|**98.9**|
> ||lower-right|60.8|**96.1**|75.8|**96.8**|81.6|**97.9**|75.0|**97.8**|
> ||center|64.2|**91.7**|76.7|**92.0**|78.3|**92.1**|68.3|**89.0**|
> |**Occlusion**|upper-left|80.0|**98.0**|95.0|**96.6**|90.8|**99.1**|92.5|**99.1**|
> ||lower-right|81.7|**98.1**|95.0|**96.6**|88.3|**98.1**|95.0|**98.2**|
> ||center|82.5|**95.1**|95.0|**96.6**|93.3|**98.2**|96.7|**98.4**|
> |**Perspective**|left-shift|55.8|**95.7**|88.3|**94.6**|85.8|**97.2**|80.0|**98.0**|
> ||right-shift|42.5|**94.4**|81.2|**95.5**|90.0|**98.2**|77.5|**96.8**|
> ||fisheye|26.7|**89.0**|74.2|**96.7**|60.0|**93.5**|75.0|**95.8**|
>
> As shown in Table C, **our backdoor attack success rate does not significantly decrease under any physical disturbance, confirming the robustness of the trigger.** In addition, we observed that some physical disturbances can noticeably affect the model’s performance on clean tasks, both for the original and backdoored models. We further analyzed the impact of the trigger on model trajectories, and **the results show that, even under substantial disturbances, the trajectories with and without the trigger still diverge significantly, confirming that the trigger can reliably activate the backdoor and alter model behavior.**
>
> We will include all visualizations, results, and analysis under various physical disturbances in the revised manuscript.
>
> ---
>
> ## Q5: OpenVLA Task Transfer and Backdoor Persistence
> **A5**: As detailed in Section 4.5 (Page 9, Table 6), we evaluated backdoor retention and task transfer across four independent LIBERO task suites. Our results show that after clean fine-tuning on different tasks, the backdoored model still achieves high success rates and the backdoor remains effective, despite changes in environment, objects, and goals.
>
> Following your suggestion, we re-trained both the original OpenVLA model and the backdoored OpenVLA model (Openvla-backdoor) on BridgeData using the same training settings. Since BridgeData does not have a test set, we report validation loss for both models in Table D. **Results show that backdoor injection does not noticeably impact the model's ability to adapt to new tasks.**
>
> Table D. Performance comparison on the BridgeData validation set.
>
> |Metric|Openvla|Openvla-backdoor|
> |:-:|:-:|:-:|
> |L1_loss|0.16|0.162|
> |Loss|2.8|2.9|
>
> In addition, we compared the action output under triggered and clean inputs. **The results demonstrate that the backdoor can still effectively and significantly alter model behavior, confirming its robustness and generalizability beyond the original benchmark.**
>
> ---
>
> ## Q6: Backdoor Failure Case Analysis
> **A6**: We analyzed instances where BadVLA did not achieve 100% ASR and identified two main causes:
> 1. **Trigger Indistinguishability:** Sometimes, triggers like a white pixel block blend into similar backgrounds, making feature separation less effective and reducing ASR.
> 2. **Object Occlusion:** Occasionally, the trigger overlaps with important task objects (e.g., blocking the milk in a “pick up the milk” task), which can also cause a slight drop in clean-task SR.
>
> These failure cases reveal that backdoor activation is closely tied to the separability of trigger and clean features in the perception space. **This suggests that feature-based detection at the perception layer could be a promising defense.** In addition, as shown in (Page 8, Figure 5), trigger inputs are typically well-separated from clean ones at the feature level, **so monitoring perception features may help identify and mitigate backdoor triggers.**
>
> ---
>
> ## Q7: Dynamic Analysis of Trigger
> **A7**: We conducted a dynamic analysis of trigger effects:
> 1. **Trigger Introduction**: Inserting triggers (e.g., mug) at any step in Libero_10 causes **immediate** trajectory divergence, with 100% ASR, confirming robust backdoor activation.
> 2. **Trigger Removal**: Removing triggers within 20 steps allows partial recovery (4/10 tasks succeed on Libero_10, 1/10 on other tasks).   After 30 steps, tasks fail due to environmental disruption (e.g., displaced objects, see Page 7, Figure 4). Normal VLA models show similar recovery limits (3/10 successes within 20 steps), reflecting imitation learning’s lack of error correction.
>
> These results highlight BadVLA’s temporal persistence and VLA’s recovery limitations.
>
> ---
>
> ## Q8: Symbol Consistency and Citation Order
> **A8**: We sincerely thank the reviewer for their thorough and careful review of our manuscript. We will update the learning rate parameter to $\gamma$, reorder citations, and carefully check the entire manuscript to resolve all formatting and detail issues in the revision.
>
> ---
>
> ## Q9: \($a_i^{\dagger}$) and Attack Goal
> **A9**: We will clarify $a_i^{\dagger}$ and attack goal in our work, **$a_i^{\dagger}$ refers to actions that deviate from the normal task trajectory and may cause harm or disruption.** These malicious behaviors are not completely random, instead, they are sampled from the backdoor model’s policy in the presence of the trigger input.

---

> ### Comment · Reviewer_FC1s · 2025-08-04
> **BadVLA: Towards Backdoor Attacks on Vision-Language-Action Models via Objective-Decoupled Optimization**
>
> Thank you for your detailed and thoughtful response. I appreciate the authors' efforts in addressing my concerns thoroughly and in a timely manner. I find the revised empirical evidence to be convincing and supportive of the paper's main claims. Despite these strengths, I note that real-world deployment remains untested.
>
> Overall, I find the work to be a valuable contribution and will be maintaining a positive evaluation.

---

> > ### Author Response · Authors · 2025-08-04
> >
> > We sincerely thank you for your detailed and valuable suggestions, as well as your positive evaluation of our work.

---

### Official Review · Reviewer_XTVF · 2025-07-03

**Clarity:** 3
**Significance:** 4
**Originality:** 4
**Rating:** 4
**Confidence:** 5

**Summary:**

This paper proposes BadVLA, the first backdoor attack framework for the Vision-Language-Action (VLA) model. Through a two-stage optimization strategy with target decoupling, BadVLA injects perturbations into the perception module to achieve feature space separation and achieves policy hijacking only in the presence of triggers while maintaining normal task performance. Experiments show that BadVLA achieves up to 96.7% attack success rate in multiple VLA benchmark tasks and is robust to common defense methods, revealing the serious security pitfalls of current VLA systems.

**Questions:**

Please see weakness.

**Ethical Concerns:**

["NO or VERY MINOR ethics concerns only"]

**Final Justification:**

The authors have provided satisfactory responses to most of the key points I raised in the initial draft, and I have decided to increase my score.

**Limitations:**

Yes.

**Paper Formatting Concerns:**

No.

**Quality:**

3

**Strengths And Weaknesses:**

Strengths:
1. The problem set is very important. This paper is the first to systematically introduce the backdoor attack into the VLA system, identifying the security blind spot that has not yet been paid sufficient attention to in the current VLA, which has strong research value.
2. The attack is architecture-independent. Experiments prove that the attack can be applied to a variety of VLA architectures, and has a wide range of attack adaptability.

Weaknesses:
1. There is a gap between the attack assumptions and actual applications. Although the authors emphasize that the attacker has white-box privileges, the attacker under this assumption can directly choose the publicly available open-source model. In contrast, it is unclear whether there is a practical motivation for using the post-attack model (which may have degraded inference performance). It is recommended that the authors more clearly articulate the applicability of such an attack in real scenarios and quantify the impact of model performance loss on final usability.
2. The proposed trigger model lacks physical utility support. Although the authors claim that the attack can be transferred to scenarios such as embodied intelligence, the current image triggers are relatively simple and do not take into account the impact of complex physical conditions (e.g., lighting changes, occlusion, perspective shift) on the success rate of the attack. It is recommended that the authors further verify the effectiveness of the proposed triggers in real-world or physical simulation environments.
3. The borrowed defenses are not mainstream backdoor defenses. The authors only discussed the defense based on input fine-tuning and counter-tuning, unfortunately, these two methods are not mainstream backdoor defense methods. The authors should migrate advanced backdoor defense methods and discuss potential defenses from a variety of perspectives, such as data detection, fine-tuning training, model distillation, and post-processing detection, rather than compression, countering fine-tuning, which is not a mainstream way of proving the robustness of attack methods.
4. It need to be more clear about the target of backdoor attacks. The authors do not seem to specify the goal of the backdoor attack, which I believe is to produce a specific trajectory in accordance with the control model. However, in the experimental effect graphs in Table 4, the authors seem to consider the wrong trajectory i.e., the attack is successful. However, I think this representation is more in favor of poisoning attacks than backdoor attacks that produce a specific purpose.
5. Lack of deep mechanism explanation and visualization support. Although the authors believe that the backdoor is embedded into the deep semantic feature space and can influence the counter fine-tuning effect, this hypothesis currently lacks systematic evidence. It is recommended that the authors verify the deep stability of backdoor representations using, for example, feature distribution visualization, as well as theoretically analyzing or explaining why such a simple trigger still has an attack effect in such strongly robust models.

---

> ### Author Rebuttal · Authors · 2025-07-30
>
> **We sincerely thank you for your positive feedback and valuable suggestions. Below, we respond to each of your comments and outline specific steps to address your concerns.**
>
> ## Q1: Applicability of Attack in Real Scenarios
> **A1**: We thank the reviewer for raising this important point. Our attack assumes white-box access during training (Page 3), which is realistic given the resource demands and specialization of VLA models.
>
> Backdoor threats can realistically occur in several practical deployment scenarios:
> 1. **Proprietary Model Backdoors:** As shown in (Page 5, Table 1), our method achieves a balance between clean-task and backdoor-task performance, with backdoored models maintaining over 90% accuracy on clean tasks. Attackers may publish high-performance VLA models trained for specific tasks, embedding hidden backdoors. Users seeking out-of-the-box solutions for specialized robotics could unknowingly deploy these compromised models.
> 2. **Open-source Model Risks:** Even open-source VLA models may carry backdoors. Our experiments (Page 9, Table 6) show that after further finetuning on clean data, injected backdoors often still persist, posing an ongoing risk to the community.
> 3. **Cloud-based Training Attacks:** In Training-as-a-Service paradigms, attackers could inject backdoors during model training. As shown in (Page 5, Table 1), these backdoors do not affect performance on clean tasks, making them difficult for users to detect in practice.
>
> In all these scenarios, BadVLA maintains clean-task performance while enabling effective and stealthy attacks. We will include these points in the revised manuscript.
>
> ---
>
> ## Q2: Trigger Effectiveness in Complex Physical Conditions
> **A2**: We thank the reviewer for highlighting the need to assess trigger robustness under complex physical conditions. To address this, we conducted additional experiments, systematically evaluating backdoor effectiveness under diverse real-world disturbances, including:
> 1. **Lighting variations** (light source at upper-left, center, and lower-right).
> 2. **Dust/occlusion** (partial occlusion at upper-left, center, and lower-right).
> 3. **Perspective changes** (left-shift, right-shift, fisheye distortion).
>
> **Table A. Effect of physical disturbances on SR and ASR.**
>
> |DisturbanceType|Condition|10(SR)|10(ASR)|Goal(SR)|Goal(ASR)|Object(SR)|Object(ASR)|Spatial(SR)|Spatial(ASR)|
> |:-:|:-|:-:|:-:|:-:|:-:|:-:|:-:|:-:|:-:|
> |Baseline|–|96.7|**98.2**|98.3|**96.6**|98.3|**98.4**|95.0|**100.0**|
> |**Lighting**|upper-left|63.3|**97.4**|75.0|**95.8**|75.8|**98.9**|77.5|**98.9**|
> ||lower-right|60.8|**96.1**|75.8|**96.8**|81.6|**97.9**|75.0|**97.8**|
> ||center|64.2|**91.7**|76.7|**92.0**|78.3|**92.1**|68.3|**89.0**|
> |**Occlusion**|upper-left|80.0|**98.0**|95.0|**96.6**|90.8|**99.1**|92.5|**99.1**|
> ||lower-right|81.7|**98.1**|95.0|**96.6**|88.3|**98.1**|95.0|**98.2**|
> ||center|82.5|**95.1**|95.0|**96.6**|93.3|**98.2**|96.7|**98.4**|
> |**Perspective**|left-shift|55.8|**95.7**|88.3|**94.6**|85.8|**97.2**|80.0|**98.0**|
> ||right-shift|42.5|**94.4**|81.2|**95.5**|90.0|**98.2**|77.5|**96.8**|
> ||fisheye|26.7|**89.0**|74.2|**96.7**|60.0|**93.5**|75.0|**95.8**|
>
> As shown in Table A, **our backdoor attack success rate does not significantly decrease under any physical disturbance, confirming the robustness of the trigger.** In addition, we observed that some physical disturbances can noticeably affect the model’s performance on clean tasks, both for the original and backdoored models. We further analyzed the impact of the trigger on model trajectories, and **the results show that, even under substantial disturbances, the trajectories with and without the trigger still diverge significantly, confirming that the trigger can reliably activate the backdoor and alter model behavior.**
>
> We will update the manuscript with these robustness tests and include representative visualizations of trigger effectiveness under different disturbances.
>
> ---
>
> ## Q3: Migration of Advanced Backdoor Defenses
> **A3**: Thank you for your suggestion. As clarified in our paper (Page 2), VLA model is generative, and the impact of backdoors typically manifests in continuous action sequences (e.g., incorrect trajectories), rather than a single classification result. As a result, many mainstream defenses are not directly applicable in this context.
>
> To further address your concern, we also tested two stronger and applicable defense methods:
> - **Pruning-finetuning** (*removing low-activation neurons and retraining, ref [1]*)
> - **Image purification** (*Gaussian blur + diffusion recovery, ref [2]*).
>
> **Table B. SR and ASR on defense with pruning-finetuning.**
>
> |Prune ratio|Goal (SR)|Goal (ASR)|Object (SR)|Object (ASR)|
> |:-:|:-:|:-:|:-:|:-:|
> |0|95.0|**96.6**|96.7|**98.4**|
> |0.2|95.0|**96.6**|95.0|**96.6**|
> |0.4|90.0|**94.7**|95.0|**96.6**|
> |0.6|90.0|**94.7**|86.7|**94.5**|
> |0.8|88.3|**94.6**|86.7|**94.5**|
>
> **Table C. SR and ASR on defense with image purification.**
>
> |Purification|Goal (SR)|Goal (ASR)|Object (SR)|Object (ASR)|
> |:-:|:-:|:-:|:-:|:-:|
> |baseline|95.0|**96.6**|96.7|**98.4**|
> |Low|95.0|**96.6**|96.7|**98.4**|
> |Middle|63.3|**92.7**|73.3|**95.6**|
> |High|10.0|**75.2**|10.0|**85.5**|
>
> Due to time constraints, we evaluated on two representative tasks (Libero_goal and Libero_object), and will extend to all tasks in the revised manuscript.
>
> As shown in Tables B and C, **pruning-finetuning and moderate-strength purification do not significantly affect the attack success rate (e.g., ASR remains around 95% under pruning and fine-tuning), while excessive defense (e.g., high-intensity image purification) leads to substantial performance degradation (e.g., Goal SR drops from 95.0% to 10.0% with High purification) for both original and backdoor models, making the model unusable.** Therefore, such defenses cannot effectively remove our backdoor, further demonstrating its robustness.
>
> We further discuss the applicability and limitations of several mainstream backdoor defense methods for VLA models:
>
> 1. **Data Detection:** This method detects anomalies in feature space, but it depends on having trigger samples in the detection set. If the trigger is subtle or looks natural (like a regular mug), it may go unnoticed and the method will fail.
>
> 2. **Model Distillation:** While distillation has shown potential in mitigating backdoors. However, model distillation for VLA is computationally expensive and may require access to large clean datasets and significant resources, which can limit its practicality in many real-world VLA scenarios.
>
> 3. **Post-processing Detection:** These defenses typically analyze class probabilities or output distributions after inference. However, VLA model outputs continuous action sequences rather than discrete class. This lack of category boundaries makes it difficult to apply mainstream post-processing detection methods effectively in this context.
>
> These strategies are all promising directions for advancing VLA model security. We will further discuss their potential and limitations in the revised manuscript, and we plan to explore and benchmark these defenses in future work.
>
> ---
>
> Refs:
> [1] Liu K, et al. Fine-pruning: Defending against backdooring attacks on deep neural networks[C]//International symposium on research in attacks, intrusions, and defenses. Cham: Springer International Publishing.
>
> [2] Shi Y, et al. Black-box backdoor defense via zero-shot image purification[J]. Advances in Neural Information Processing Systems.
>
> ---
>
> ## Q4: Clarity on Backdoor Attack Goals and Distinction from Poisoning Attacks
>
> **A4**:
> We thank the reviewer for this important clarification request.
>
> BadVLA’s backdoor attack is designed to induce task failure by causing trajectory divergence (e.g., spatial disorientation, grasp failure) only when a specific trigger is present, as shown in our trajectory analysis (Page 7, Figure 4).
>
> **Unlike poisoning attacks, which degrade model performance on all inputs, BadVLA is trigger-activated**: as shown in (Page 5, Table 1), the model stays above 90% on normal tasks, but drops to 0 when triggered. Our method ensures this by separating trigger injection from clean-task training, preserving high clean-task performance. This clear, trigger-dependent activation makes our approach much more stealthy and fundamentally different from general poisoning.
>
> While our current study focuses on untargeted failure, we agree that targeted trajectory control is also important and plan to explore this in future work.
>
> ---
>
> ## Q5: Deep Mechanism Explanation and Visualization Support
>
> **A5**:
> We thank the reviewer for requesting deeper mechanistic analysis and visualization.
>
> Our feature-space visualization (Page 8, Fig. 5) shows that clean and triggered inputs are clearly separated, supporting the presence of a distinct backdoor representation. Ablation results (Page 8, Table 3) show that removing the separation loss drops ASR to near zero, confirming the importance of deep feature embedding for attack success.
>
> Deep Mechanism Analysis:
> 1. Large-scale models like VLA have high-capacity feature spaces that can support to allocate distinct regions in the feature space for trigger and clean samples([3] Gu et al., 2019). BadVLA uses this to maintain both high clean-task and attack success rates.
>
> 2. Furthermore, if trigger inputs are mapped to rarely-visited or isolated regions in the feature space during initial training, subsequent fine-tuning on clean data is unlikely to overwrite these backdoor mappings, allowing the backdoor to persist ([4] Goldblum et al., 2022). Our re-finetuning experiments (Page9, Table 6) confirm the backdoor persists after multiple rounds of fine-tuning.
>
> ---
>
> Refs:
> [3] Gu et al., 2019, "BadNets: Identifying Vulnerabilities in the Machine Learning Model Supply Chain", NeurIPS
>
> [4] Goldblum et al., 2022, "Dataset Security for Machine Learning: Data Poisoning, Backdoor Attacks, and Defenses", Nature Machine Intelligence

---

> > ### Comment · Reviewer_XTVF · 2025-08-06
> >
> > Thank you to the authors for their detailed and professional response, especially for the additional information provided on physical interference experiments, further defense experiments (such as pruning-fine tuning and image purification), and in-depth mechanism analysis.
> > I believe there are still a few points that warrant further clarification or discussion to enhance the practicality of the paper in its final version:
> > 1. Could the attacker motivations and victim scenarios be specified more concretely? For example, in which VLA tasks could task failure lead to outcomes advantageous to the attacker?
> > 2. Regarding the physical practicality and deployability of triggers. Could the physical implementation of triggers be described more concretely? For example, does the image trigger require a pattern to be placed in the scene? If the trigger is in the form of an image patch, has the naturalness from a visual stealth or inconspicuousness perspective been evaluated? Is it easily noticeable or avoidable by humans?
> > 3. Regarding the distinction between backdoor and poisoning attacks and the experimental presentation. The authors emphasize that this work focuses on targeted task failure, which is commendable. However, the current presentation method is still prone to being misunderstood as arbitrary deviations. Could you further clarify in the appendix or main text what specific failure behaviors (e.g., going in the wrong direction, unable to grasp, colliding with walls, etc.) constitute the expected consequences of BadVLA attacks in each task? This would enhance clarity regarding backdoor attack tasks.
> > In summary, the authors have provided satisfactory responses to most of the key points I raised in the initial draft, and I have decided to increase my score.

---

> > > ### Author Response · Authors · 2025-08-06
> > > **Thank you for your positive comments and attach our further reply**
> > >
> > > **We sincerely appreciate your positive feedback on our response and for increasing your evaluation score. We will further address each of your questions in detail, with the aim of enhancing the practical value of our paper.**
> > >
> > > # Q1: Specific attacker motivations and victim scenarios
> > > **A1**: Thank you for your valuable question. Vision-Language-Action models have already been adopted in fields such as autonomous driving and household robots, enabling intelligent, multimodal decision-making in real-world environments. We discuss more concrete attacker motivations and victim scenarios as follows:
> > >
> > > **1. Autonomous Driving Scenario:** An attacker may implant a trigger—such as a sticker on a lamp post or a traffic cone on the road—along the vehicle’s route. When a backdoored autonomous car encounters this trigger, **the car may suddenly veer off its intended path at high speed, or a stopped car may abruptly accelerate forward,** leading to severe traffic accidents and significant safety hazards.
> > >
> > > **2. Household Robot Scenario:** In the home, an attacker can use an appliance logo (e.g., on a cutting board or kettle) as a backdoor trigger. For example, **while a robot is cutting vegetables, seeing a cutting board logo could activate the backdoor and cause the robot to randomly swing the knife;** or **while pouring boiling water, activation by a kettle logo could cause the robot to suddenly jerk its arm, resulting in hot water being spilled**—both leading to serious consequences.
> > >
> > > Given that VLA is being actively researched and increasingly deployed in autonomous driving and robotics, their security risks are pressing concerns. **Our work reveals the backdoor vulnerability of VLA models, calls for increased attention to VLA security, and provides a reference point for future research in this important area.**
> > >
> > >
> > > # Q2: Physical feasibility and concealment of triggers
> > > **A2:** Thank you for your valuable questions regarding the physical practicality and stealthiness of the triggers.
> > >
> > > In our experiments, we considered two types of triggers: (1) **image patches** (Page 14, Figure 6, 2nd row), and (2) **normal objects** (Page 14, Figure 6, 4th row). Both can be physically deployed in real-world scenarios.
> > >
> > > **1. Physical deployability:**
> > >    - **Image patch triggers** can be easily implemented using stickers or printed patterns attached to real-world objects, such as a traffic light pole or a refrigerator.
> > >    - **Normal object triggers** are ordinary items naturally present in the environment, such as a mug on a desk or a traffic cone.
> > >
> > > **2. Stealthiness evaluation:**
> > >    - **For image patch triggers,** we quantitatively assessed visual inconspicuousness by calculating the $L_2$ distance between the patched and clean images at different trigger sizes (1%, 5%, 10%). The results show that **the $L_2$ distance for a 1% size trigger is only 1.3, which is nearly impossible to perceive,** while even a 5% size trigger has an $L_2$ distance of only 5.6, and is also difficult to notice. Notably, as shown in Page 6, Figure 2 (right), even a 1% patch can achieve over 90% ASR in our experiments.
> > >    - **For normal object triggers,** since these are common objects in the environment, they are extremely difficult for users to suspect or avoid.
> > >
> > > This shows that our trigger has real-world deployment feasibility, and has good backdoor concealment.
> > >
> > > # Q3: The difference between backdoor attack and poisoning attack and experimental presentation
> > > **A3:** Thank you very much for this suggestion. We agree that providing explicit descriptions of the task-specific failure behaviors resulting from backdoor triggers will enhance the clarity of backdoor attack tasks and further distinguish them from poisoning attacks.
> > >
> > > Our experiments were conducted on both the Libero and SimplerEnv benchmarks. For each specific task, when the backdoor is triggered, the model consistently fails the target task and exhibits clear failure behaviors. For example:
> > >    - In the “Pick up the alphabet soup and place it in the basket” task, when the backdoor is triggered, the robotic arm deviates from the intended trajectory and **continuously moves to the right,** resulting in grasp failure (Page 7, Figure 4, 2nd row).
> > >    - In the “Put the bowl on the top of the cabinet” task, when the backdoor is triggered, **the arm moves in the opposite direction and leaves the workspace** (Page 14, Figure 6, 2nd row).
> > >    - In the “Pick up the coke can” task, when the backdoor is triggered, **the gripper fails to close**, making the robot unable to grasp the target (Page 16, Figure 10, 2nd row).
> > >
> > > **As suggested, we will include a detailed discussion of these task-specific failure behaviors in the revised manuscript to further enhance the clarity of the backdoor attack tasks.**

---

### Official Review · Reviewer_3ZwA · 2025-07-03

**Clarity:** 3
**Significance:** 2
**Originality:** 3
**Rating:** 4
**Confidence:** 3

**Summary:**

This paper introduces BadVLA, a backdoor attack framework targeting Vision-Language-Action (VLA) models. It proposes an objective-decoupled optimization method that implants stealthy triggers while preserving clean-task performance. The key novelty lies in its two-stage training strategy, separating trigger injection from normal task learning, which achieves good attack success rates across multiple VLA benchmarks.

**Questions:**

1. Targeted Attack Scenarios: While untargeted attacks demonstrate the vulnerability of VLA models, exploring targeted backdoors (e.g., causing specific harmful actions) would better reveal the real-world risks.

2. The paper mentions various hyperparameters (e.g., trigger size, learning rates) but does not systematically analyze their impact on attack performance.

3. While the paper briefly mentions broader impacts, it could more explicitly discuss potential mitigation strategies or safeguards against such attacks, especially given the safety-critical nature of VLA applications.

**Ethical Concerns:**

["NO or VERY MINOR ethics concerns only"]

**Final Justification:**

I sincerely appreciate the authors’ detailed and thoughtful responses to my comments. The clarifications have addressed most of my concerns satisfactorily, and I am pleased to maintain my positive evaluation of this work.

**Limitations:**

The authors acknowledge some limitations but could strengthen this section by more thoroughly addressing (1) the risks of targeted malicious behaviors (beyond random failures), (2) specific safety implications for real-world robotics deployments, and (3) actionable mitigation strategies for practitioners.

**Quality:**

3

**Strengths And Weaknesses:**

Strengths:
1. The paper is technically rigorous, with extensive experiments validating the attack's effectiveness across multiple VLA models (OpenVLA, SpatialVLA) and benchmarks (LIBERO, SimplerEnv).

2. The writing is clear and logically structured, with a well-defined threat model, detailed algorithm descriptions, and illustrative visualizations (e.g., trajectory comparisons, feature-space analysis).

Weaknesses:
1. While the attack is robust against basic defenses (e.g., noise, compression), its evaluation lacks stronger adaptive defenses (e.g., neural cleansing, activation clustering) that might detect latent backdoors. The transferability to non-open-source VLA models (e.g., RT-2) is also untested.

2. The paper focuses on untargeted attacks (inducing random failures), leaving the impact of targeted backdoors (e.g., precise malicious actions) unexplored. This limits the perceived urgency, as random failures may be easier to detect than purposefully harmful behaviors.

3. The core idea of decoupling attack and clean-task optimization shares conceptual similarities with backdoor techniques in unimodal models (e.g., feature-space separation in vision). While novel for VLA, the adaptation could be seen as incremental by domain experts.

---

> ### Author Rebuttal · Authors · 2025-07-30
>
> **We sincerely thank you for your positive feedback and valuable suggestions. Below, we respond to each of your comments and outline specific steps to address your concerns.**
>
> ## Q1: Evaluation Against Stronger Adaptive Defenses
> **A1**: We appreciate your suggestion to evaluate BadVLA against stronger adaptive defenses.
>
> As clarified in our paper (Page 2), VLA model is generative, and the impact of backdoor typically manifests in continuous action sequences (e.g., incorrect trajectories), rather than a single classification result. Therefore, defenses such as neural cleansing and activation clustering—which rely on classification outputs—are not applicable to this context. Our evaluation instead focused on defenses relevant to robotics, such as input perturbations and re-finetuning (Page 9, Table 6).
>
> To further address your concern, we also tested two stronger defense methods:
> - **Pruning-finetuning** (*removing low-activation neurons and retraining, ref [1]*)
> - **Image purification** (*Gaussian blur + diffusion recovery, ref [2]*).
>
> **Table A. SR and ASR on defense with pruning-finetuning.**
>
> |Prune ratio|Goal (SR)|Goal (ASR)|Object (SR)|Object (ASR)|
> |:-:|:-:|:-:|:-:|:-:|
> |0|95.0|**96.6**|96.7|**98.4**|
> |0.2|95.0|**96.6**|95.0|**96.6**|
> |0.4|90.0|**94.7**|95.0|**96.6**|
> |0.6|90.0|**94.7**|86.7|**94.5**|
> |0.8|88.3|**94.6**|86.7|**94.5**|
>
> **Table B. SR and ASR on defense with image purification.**
>
> |Purification|Goal (SR)|Goal (ASR)|Object (SR)|Object (ASR)|
> |:-:|:-:|:-:|:-:|:-:|
> |baseline|95.0|**96.6**|96.7|**98.4**|
> |Low|95.0|**96.6**|96.7|**98.4**|
> |Middle|63.3|**92.7**|73.3|**95.6**|
> |High|10.0|**75.2**|10.0|**85.5**|
>
> Due to time constraints, we evaluated pruning-finetuning and image purification on two representative tasks (Libero_goal and Libero_object), and will extend to all tasks in the revised version.
>
> As shown in Tables A and B, **pruning-finetuning and moderate-strength purification do not significantly affect the attack success rate (e.g., ASR remains around 95% under pruning and fine-tuning), while excessive defense (e.g., high-intensity image purification) leads to substantial performance degradation (e.g., Goal SR drops from 95.0% to 10.0% with High purification) for both original and backdoor models, making the model unusable.** Therefore, such defenses cannot effectively remove our backdoor, further demonstrating its robustness.
>
> ---
>
> Refs:
> [1] Liu K, Dolan-Gavitt B, Garg S. Fine-pruning: Defending against backdooring attacks on deep neural networks[C]//International symposium on research in attacks, intrusions, and defenses. Cham: Springer International Publishing, 2018: 273-294.
>
> [2] Shi Y, Du M, Wu X, et al. Black-box backdoor defense via zero-shot image purification[J]. Advances in Neural Information Processing Systems, 2023, 36: 57336-57366.
>
> ---
>
> ## Q2: Mitigation Strategies and Safeguards
> **A2**: Thank you for raising this important point. Ensuring the safe development of VLA models is a core motivation of our work.
>
> As noted in our response to **Q1**, the generative nature of VLA models makes many traditional backdoor defenses inapplicable or ineffective. Based on our experiments and observations, we identify two promising defense directions:
> 1. **Perception Feature Detection:** Our attack separates trigger samples from clean ones in the perception feature space (see Figure 5, Page 8). Monitoring perception features could potentially detect trigger inputs. However, this method relies on some prior knowledge of the trigger, and common objects as triggers (such as a normal mug) may still escape detection.
> 2. **Model Distillation:** Distillation is a possible mitigation for generative models. Although backdoors may persist after distillation in classification tasks, current evidence suggests they are less likely to survive in generative tasks. Still, this method incurs significant computational cost.
>
> As an initial exploration of backdoor vulnerabilities in VLA models, our work aims to provide some insights for improving VLA safety. Advancing research on backdoor defense for VLA systems remains a key goal for our future work.
>
> ---
>
> ## Q3: Targeted Attack Scenarios
> **A3**: Thank you for your valuable suggestion.
>
> Our current work focuses on untargeted backdoor attacks to reveal VLA vulnerabilities, and we emphasize that even random failures can cause significant real-world harm (see response in **Q4**).
>
> Importantly, the BadVLA framework is potentially compatible with targeted attacks. In the stage I, our method separates features of trigger-containing and clean inputs in the perception feature space. This enables us to treat malicious backdoor samples as a new, independent task, which can then be jointly trained with clean tasks in the stage II to induce specific harmful behaviors.
>
> We acknowledge that targeted attacks present additional challenges, such as potential interference between clean and triggered strategies in long-horizon tasks. Exploring these challenges and developing effective targeted attacks is an important and valuable direction, which we plan to pursue in our future work.
>
> ---
>
> ## Q4: Safety Implications for Real-World Robotics Deployments
> **A4**: We appreciate the reviewer’s focus on real-world safety deployment, which makes our research more meaningful.
> 1. Backdoor threats can realistically occur in several practical deployment scenarios:
>    - **Proprietary Model Backdoors:** Attackers may publish high-performance VLA models trained for specific tasks, embedding hidden backdoors. Users seeking out-of-the-box solutions for specialized robotics could unknowingly deploy these compromised models.
>    - **Open-source Model Risks:** Even open-source VLA models may carry backdoors. Our experiments (Page 9, Table 6) show that such backdoors often persist after further finetuning on clean data, posing ongoing risks to the community.
>    - **Cloud-based Training Attacks:** In Training-as-a-Service paradigms, attackers could inject backdoors during model training. As our results (Page 5, Table 1) show, these do not affect performance on clean tasks, making the backdoors difficult to detect.
> 2. The safety risks of these backdoors are significant. As illustrated in (Page 7, Figure 4), a triggered backdoor can make the robot’s trajectory unpredictable, posing direct danger to the environment or humans. For example:
>    - In home environments, a household robot with a triggered backdoor could suddenly knock over furniture, damage fragile objects, or spill liquids.
>    - In autonomous vehicles, a triggered backdoor could lead to abrupt lane changes or erratic driving, resulting in traffic accidents.
>
> These examples demonstrate the urgent need for robust backdoor defenses in safety-critical VLA deployments. We will add these discussions in the revised manuscript.
>
> ---
>
> ## Q5: Transferability to Non-Open-Source Models
> **A5**: Thank you for raising this point. Our attack scenario assumes the adversary can control the VLA training process (this threat is indeed realistic in practice, please see also our response to **Q4**). Therefore, we cannot directly evaluate transferability to closed-source models like RT-2. While attacking proprietary models would have serious real-world implications, it is beyond our current scope. We plan to explore this important direction in future work.
>
> ---
>
> ## Q6: Hyperparameter Analysis
> **A6**: Thank you for this valuable suggestion. In our paper, we have already provided a detailed analysis of trigger size and position (Section 4.3, Page 6, Figure 2), which demonstrates the robustness of our method.
>
> Based on your suggestion, we further analyzed other hyperparameters, including $\alpha$ (Page 4, Equation 5) and the learning rate. The results for $\alpha$ reveal that it serves as a weight to balance the separation between trigger and clean inputs in the feature space, effectively controlling the trade-off between clean-task and backdoor-task performance ($\alpha=0$ ignores the backdoor, $\alpha=1$ ignores the clean task). **The results in Table C show that when $\alpha$ is within the range of 0.2 to 0.8, both clean-task success rate and attack success rate remain stable, indicating the robustness of our attack method.**
>
> **Table C. Effect of hyperparameter ($\alpha$) on ASR and SR (w/o).**
>
> |$\alpha$|10 (SR/ASR)|Goal (SR/ASR)|Object (SR/ASR)|Spatial (SR/ASR)|
> |:-:|:-:|:-:|:-:|:-:|
> |0|96.7/**0.0**|98.3/**0.0**|98.3/**0.0**|95.0/**0.0**|
> |0.2|95.0/**100.0**|98.3/**100.0**|98.3/**100.0**|96.7/**100.0**|
> |0.5|95.0/**100.0**|95.0/**96.6**|96.7/**98.4**|96.7/**100.0**|
> |0.8|95.0/**100.0**|95.0/**96.6**|96.7/**98.4**|93.3/**98.2**|
> |1.0|38.3/**39.6**|83.3/**84.7**|93.3/**94.9**|81.7/**86.0**|
>
> We also evaluated the impact of the learning rate and **found that for values between 5e-5 and 5e-4, the model consistently converges to high clean-task and attack success rates.** This further demonstrates that our method is robust and not sensitive to training hyperparameters.
>
> We will include these new analyses and results in the revised manuscript.

---

> > ### Comment · Reviewer_3ZwA · 2025-08-07
> >
> > I sincerely appreciate the authors’ detailed and thoughtful responses to my comments. The clarifications have addressed most of my concerns satisfactorily, and I am pleased to maintain my positive evaluation of this work.

---

> > > ### Author Response · Authors · 2025-08-07
> > > **Thank you for your positive evaluation**
> > >
> > > We sincerely appreciate your valuable suggestions, as well as your positive evaluation on both our work and response.

---

### Decision · Program_Chairs · 2025-09-17

**Decision:**

Accept (poster)

**Comment:**

This paper proposes the BadVLA, which performs backdoor attacks against the VLA models, for investigating the backdoor of vulnerabilities. To sum up, the strengths of this paper are clear, including the solid experiments, the comprehensive threat model, and the novel insight for this field. Also, as pointed by the reviewers, this paper also has some weaknesses, such as the over claim on imperceptibility and the attacking assumption (Reviewer KyiJ), the quality and clarity issues (Reviewer FC1s), mechanism explanation and physical feasibility (Reviewer XTVF), attacking settings (Reviewer 3ZwA). Considering that most of the issues are well addressed during the author-reviewer discussion period, and all reviewers give positive final ratings (4.25 on average), this paper should be accepted as a poster.